Resource

# MISATO: machine learning dataset of protein–ligand complexes for structure-based drug discovery

Till Siebenmorgen [1,2,9], Filipe Menezes [1,2,9], Sabrina Benassou[3], Erinc Merdivan[4], Kieran Didi [5], André Santos Dias Mourão[1,2], Radosław Kitel[6], Pietro Liò[5], Stefan Kesselheim [3], Marie Piraud[4], Fabian J. Theis [4,7,8], Michael Sattler [1,2] & Grzegorz M. Popowicz [1,2] ✉

Large language models have greatly enhanced our ability to understand biology and chemistry, yet robust methods for structure-based drug discovery, quantum chemistry and structural biology are still sparse. Precise biomolecule–ligand interaction datasets are urgently needed for large language models. To address this, we present MISATO, a dataset that combines quantum mechanical properties of small molecules and associated molecular dynamics simulations of ~20,000 experimental protein–ligand complexes with extensive validation of experimental data. Starting from the existing experimental structures, semi-empirical quantum mechanics was used to systematically refine these structures. A large collection of molecular dynamics traces of protein–ligand complexes in explicit water is included, accumulating over 170 μs. We give examples of machine learning (ML) baseline models proving an improvement of accuracy by employing our data. An easy entry point for ML experts is provided to enable the next generation of drug discovery artificial intelligence models.

In recent years, artificial intelligence (AI) predictions have revolutionized many fields of science. In structural biology, AlphaFold2 (ref. 1) predicts accurate protein structures from amino-acid sequences only. Its accuracy nears state-of-the-art experimental data. The success of AlphaFold2 is made possible due to a rich database of nearly 200,000 protein structures that have been deposited and are available in the Protein Data Bank (PDB)[2]. These structures were determined over the past decades using X-ray crystallography, nuclear magnetic resonance (NMR) or cryo-electron microscopy. Despite enormous investments,

there are still few new drugs approved yearly, with development costs reaching several billion dollars[3]. An ongoing grand challenge is rational, structure-based drug discovery (DD). Compared with protein structure prediction, this task is substantially more difficult.

In the early stages of DD, structure-based methods are popular and efficient approaches. The biomolecule provides the starting point for rational ligand search. Later, it guides optimization to optimally explore the chemical combinatorial space[4] while still ensuring drug-like properties. In silico methods that are in principle able to tackle

[1]Molecular Targets and Therapeutics Center, Institute of Structural Biology, Helmholtz Munich, Neuherberg, Germany. [2]TUM School of Natural Sciences, Department of Bioscience, Bayerisches NMR Zentrum, Technical University of Munich, Garching, Germany. [3]Jülich Supercomputing Centre, Forschungszentrum Jülich, Jülich, Germany. [4]Helmholtz AI, Helmholtz Munich, Neuherberg, Germany. [5]Computer Laboratory, Cambridge University, Cambridge, UK. [6]Faculty of Chemistry, Jagiellonian University, Krakow, Poland. [7]Computational Health Center, Institute of Computational Biology, Helmholtz Munich, Neuherberg, Germany. [8]TUM School of Computation, Information and Technology, Technical University of Munich, Garching, Germany. [9]These authors contributed equally: Till Siebenmorgen, Filipe Menezes. ✉e-mail: grzegorz.popowicz@helmholtz-munich.de

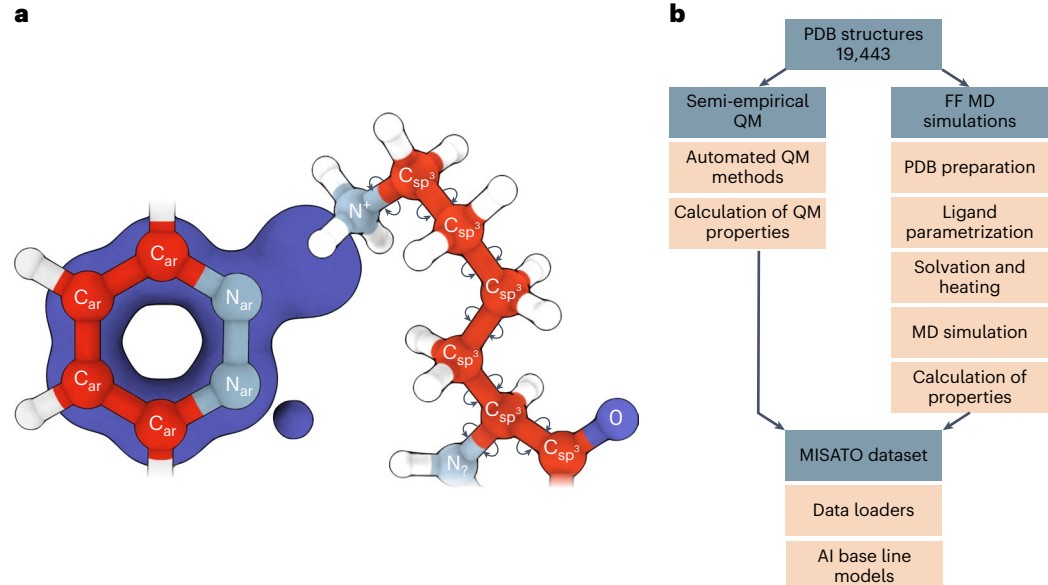

**Fig. 1 | MISATO combines QM data with MD-derived protein–ligand dynamics. a**, We provide a dataset that combines semi-empirical QM properties of small molecules with MD-simulated dynamics of the entire set of experimental protein–ligand complexes. All common errors in protein and ligand nomenclature, protonation, geometry and so on are fixed. The blue outline of the molecule describes its electronic density. **b**, An overview of the dataset and the applied protocols for semi-empirical QM and FF (force field) MD simulations including data preparation, preprocessing and AI baseline models.

structure-based DD include semi-empirical quantum mechanical (QM) methods[5], molecular dynamics (MD) simulations[6,7], docking[8] and coarse-grained simulations[9], which can also be combined to be more efficient. However, these methods either suffer from generally low precision or are computationally too expensive while still requiring substantial experimental validation. Recent examples show that classical, ball-and-stick atomistic model representations of biomolecular structures might be too inaccurate in certain situations to allow for correct predictions[10–13].

The introduction of AI into the process is still at an early stage. AI approaches are, in principle, able to learn the fundamental state variables that describe experimental data[14]. Thus, they are likely to abstract from electronic and force field-based descriptions of the protein–ligand complex. However, so far mostly simple solutions have been proposed that do not incorporate the available protein–ligand data to their full extent, such as scoring protein–ligand Gibbs free energies[15,16], ADME (absorption, distribution, metabolism and excretion) property estimation[17] or prediction of synthetic routes[18,19]. Most of these approaches are constructed using one-dimensional SMILES (simplified molecular-input line-entry system)[20,21] and only a few attempts have been made to properly tackle three-dimensional (3D) biomolecule–ligand data[22–24].

Several databases are available that contain raw experimental structures of protein–ligand complexes, usually extracted from the PDB (for example, PDBbind[25], bindingDB[26], Binding MOAD[27], Sperrylite[28]). Only recently a database of MD-derived traces of protein–ligand structures was reported[29,30]. Despite these efforts, so far no AI model has been proposed that convincingly addresses the rational DD challenge in the way that AlphaFold2 answered the protein structure prediction problem[31,32].

In addition to DD, the structure-based AI models are useful for biomolecule structure analysis and quantum chemistry. However, they are severely hindered by several factors: neglecting the conformational flexibility (dynamics and induced fit upon binding); entropic considerations; inaccuracies in the deposited structural data (incorrect atom types due to missing hydrogen atoms, incorrect evaluation of functional group flexibility, inconsistent geometry restraints, fitting errors); chemical complexity (for example, non-obvious protonation states); overly simplified atomic properties; highly complex energy landscapes in molecular recognition by their targets. Attempts to train AI models currently require inferring this missing information implicitly. The limited number of publicly available protein–ligand structures (~20,000) and lack of thermodynamic data cause this inference to fail. This is preventing structure-based models from producing groundbreaking results[31,32].

Here, we propose a protein–ligand structural database, MISATO (molecular interactions are structurally optimized) that is based on experimental protein–ligand structures. We show that the database helps to better train models across fields related to DD and beyond. This includes quantum chemistry, general structural biology and bioinformatics. We provide quantum-chemical-based structural curation and refinement, including regularization of the ligand geometry. We augment this database with missing dynamic and chemical information, including MD on a timescale allowing the detection of transient and cryptic states for certain systems. The latter are very important for successful drug design[33]. Thus, we supplement experimental data with the maximum number of physical parameters. This eases the burden on AI models to implicitly learn all this information, allowing focus on the main learning task. The MISATO database provides a user-friendly format that can be directly imported into machine learning (ML) codes. We also provide various preprocessing scripts to filter and visualize the dataset. Example AI baseline models are supplied for the calculation of quantum chemical properties (chemical hardness and electron affinity), for binding affinity calculation and for the prediction of protein flexibility or induced-fit features to simplify adoption. The QM, MD and AI baseline models are validated extensively on experimental data. We wish to transform MISATO into an ambitious community project with vast implications for the whole field of DD.

## Results

### MISATO dataset

The basis for MISATO (Fig. 1) is the 19,443 protein–ligand structures from PDBbind[25]. These structures were experimentally determined over the past decades and represent a diverse set of protein–ligand complexes for which experimental affinities are available. In the context of AI for DD it is of utmost importance to train the models on a dataset

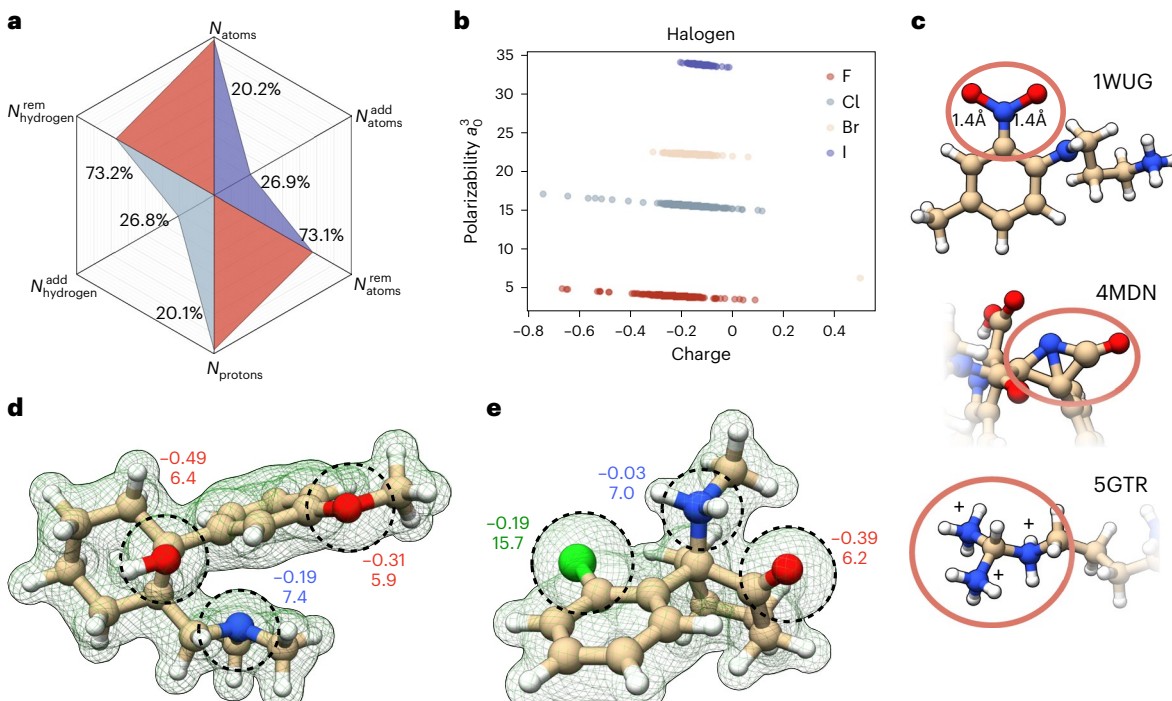

**Fig. 2 | Changes applied to the PDBbind database based on our quantum chemical protocol. a**, Statistical overview of changes introduced by our optimization protocol. $N_{atoms}$ corresponds to total changes in the atom count when compared with the source database. In most cases atoms were removed $N_{atoms}^{rem}$; in only 27% of cases was the number of atoms increased, $N_{atoms}^{add}$. Similar considerations apply to protons—light blue; $N_{protons}$. **b**, D4 polarizability versus partial charge for all the halogens in the database. The outliers were analyzed to find possible wrong atom assignments. This was the case for the bromine atom in the lower right corner, which in reality is a boron. **c**, Examples of inconsistent structures: 1WUG contains overly elongated NO bonds; 4MDN contains a nitrogen in angular violation of VSEPR; 5GTR shows a typical problem in the protonation state. **d,e**, Calculated electronic density for ketamine (4G8H) and tramadol, respectively (dashed green lines). Dashed circles show the sizes of electronic density around selected atoms. The numbers next to these atoms represent partial charge (top) and atomic polarizability (bottom). These are electronic descriptors representing the electronic density around each center. Color and character keys: N, blue, nitrogen; S, yellow, sulfur; O, red, oxygen; C, beige, carbon; H, white, hydrogen; Cl, green, chlorine.

with the highest possible correctness and consistency, for several reasons. First, the total number of available structures is much lower than typical training sizes of other AI targets. Second, ligand association has a rather complex energy landscape during molecular recognition. Delicate deviations in the protein–ligand structures or atomic parameters can markedly impair binding. In the PDB, incorrect atom assignments and inconsistent geometries are not uncommon. More seriously, hydrogen atoms are highly sensitive to their chemical and molecular environment and are rarely experimentally accessible. All these issues have been systematically addressed in our work and are compiled in our database (Figs. 2 and 3).

### Typical limitations in structural datasets
Understanding the nature and sources of errors in structural databases is imperative for improving the quality of the underlying molecular models.

Macromolecule–ligand interaction strength, the most desired baseline parameter for DD, is unfortunately also the most inaccurate metric. The diverse experimental set-ups from experimental entropy/enthalpy determination (for example, isothermal titration calorimetry) to cellular phenotypic response are given as ligand strength. These values are not comparable and their use to train AI models is generally unreliable. To enable validation of affinity prediction we have prepared a small subset of ligands with accurately determined affinities to be used as a benchmark (Supplementary Table 1). We also tested our example model against it.

As MISATO is founded on experimental data, the two main sources of structural inaccuracies must be corrected. These are limited spatial resolution of the experimental structures and problems and biases associated with the software used for processing the molecular geometries. As well as the absence of hydrogen atoms in crystallographic structures, resolution affects the heteroatom geometry. Contracted or elongated bonds are common (Fig. 2). That is, most nitro groups we examined were heavily distorted: in the 1WUG structure[34], NO bonds are almost 17% larger than reference experimental data[35]. Another example is seen in the 4MDN structure[36], where an amide was so distorted that it explicitly violated VSEPR (valence shell electron pair repulsion) theory. Reinspection of the experimental electronic density hinted that the CÔC angle in the 4-chlorobenzyl phenyl ether moiety is also larger by almost 20° against anisole, a reference compound for that bond angle[35]. Simultaneous relaxation of the two groups leads to substantial improvement, in particular an amide group very close to reference structural values. Such errors in the heteroatom skeleton propagate further when assigning and counting hydrogen atoms. In the 5GTR structure[37], a guanidino group strongly deviates from the expected planarity. The immediate consequences are incorrect atomic hybridizations and overassignment of hydrogen atoms, with a local formal charge of +3 in a radius of one bond around the central carbon. More examples are described in Supplementary Information.

### Evaluation of the QM-based ligand curation
Employing the protocol defined in Supplementary Section 6 we modified a total of 3,930 structures, which corresponds roughly to 20% of the original database that needed substantial refinement (Fig. 2). Of these, 3,905 cases involve changes in protonation states, while changes in heteroatoms involve 97 ligands. These are predominantly the addition of model functional groups to emulate covalent binding with the protein (20) or the addition of missing hydroxyl groups to boronic acids.

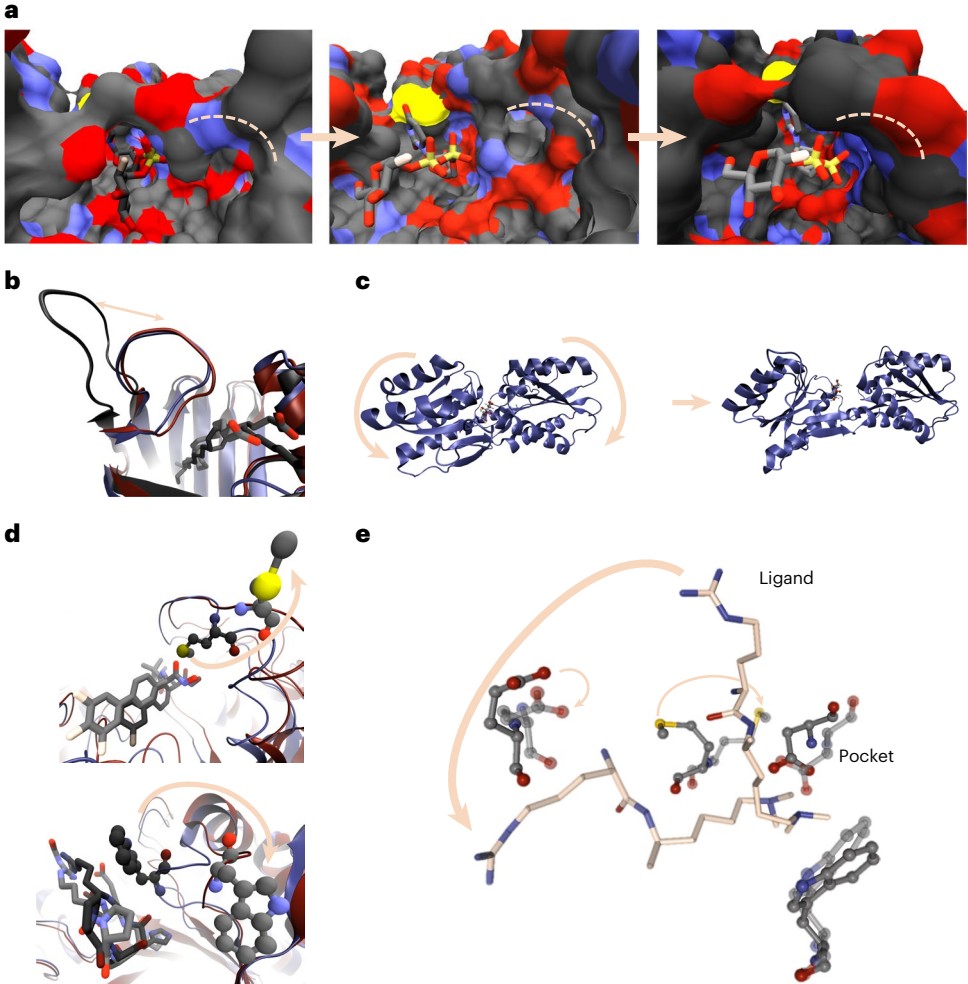

**Fig. 3 | Overview of events captured by the MD simulations in the binding pocket. a–c,** Reversible opening and closing of the binding pocket can be captured during the simulations, including cryptic binding sites. **a,** The structure of 2AM4 is shown after 2 ns (left panel), 6 ns (middle panel) and 10 ns (right panel) simulation time (fluorine in beige). **b,c,** The opening loop region (**b,** structure 2LKK) is visualized for superimposed timesteps (blue diagram, dark hue, 2 ns; black diagram, medium dark hue, 6 ns; red diagram, light hue, 10 ns). The protein pocket opens in structure 8ABP during the simulation (**c**). **d,** Protein residues at the binding site can undergo large adaptations within the simulations, indicating unstable interactions or possible switches. This is shown for a methionine residue of 4ZYZ (upper panel) and a tryptophan residue of 1WAW (lower panel). Coloring as in **b** after 2 ns and 10 ns. **e,** MD simulations captured local adaptability of the binding pocket and ligand. That is, in structure 2IG0 parts of the ligand (licorice, carbons in ivory) are quite flexible in the protein pocket (gray carbons) when comparing the first (dark hue) and the last (light hue) frames of the MD run. Color and character keys, if not indicated differently: N, blue, nitrogen; S, yellow, sulfur; O, red, oxygen; C, black, carbon; H, white, hydrogen; F, beige, fluorine; P, orange, phosphorus.

Some ligands were split into several molecules as the original structures were not binary protein–ligand complexes (one ligand): 1A0T, 1G42, 1G9D, 2L65, 3D4F and 4MNV. 1E55 is supposed to be a mixture of two entities. However, the closest contact between them is insufficient to consider them separately, but also too large for a covalent interaction. Similar considerations apply to 1F4Y, though here close intramolecular contacts are at stake. In 4AW8 we observed a substantial deformation for the published ligand, PG6. We observed that the reference affinity is related to the metal ion in the system, Zn(II), and not to PG6. The structure was consequently excluded.

As depicted in Fig. 2, the most common adjustment was the removal of hydrogen atoms from the initial PDBbind geometry. This amounts to almost 75% of the modifications. It has been pointed out that libraries such as PDBbind possess biased datasets in terms of binding configurations[31].

## QM-derived properties

We calculated several molecular and atomic properties for the ligands (Supplementary Table 2). For the former, we include electron affinities, chemical hardness, electronegativity, ionization potentials (by definition and using Koopmans' theorem), static log $P$ and polarizabilities. The latter were obtained in vacuum, water and wet octanol. Atomic properties include partial charges from different models, atomic polarizabilities, bond orders, atomic hybridizations, orbital- and charge-based reactivity (Fukui) indices and atomic softness. Reactivity indices and atomic softness are derived for interactions with electrophiles, nucleophiles and radicals. Finally, we also provide tight-binding electronic densities for all ligands. Partial charges were calculated at several levels, as these are somewhat method-sensitive quantities. AM1 charges are usually the starting point for charge-correcting schemes to be used in MD simulations. This is the case for AM1-BCC[38]. Taking our AM1 charges and multiplying them by 1.14 (in the case of neutral molecules) yields 1.14*CM1A-LBCC charges[39] used in OPLS-AA simulations[40]. The main advantage of the charges we provide is that these were obtained, when required, with a HOMO (highest occupied molecular orbital)–LUMO (lowest unoccupied molecular orbital) level shift to ensure convergence to sensible electronic states. Beyond MD simulations, CMx charges[41–43] have also been shown to provide good estimates

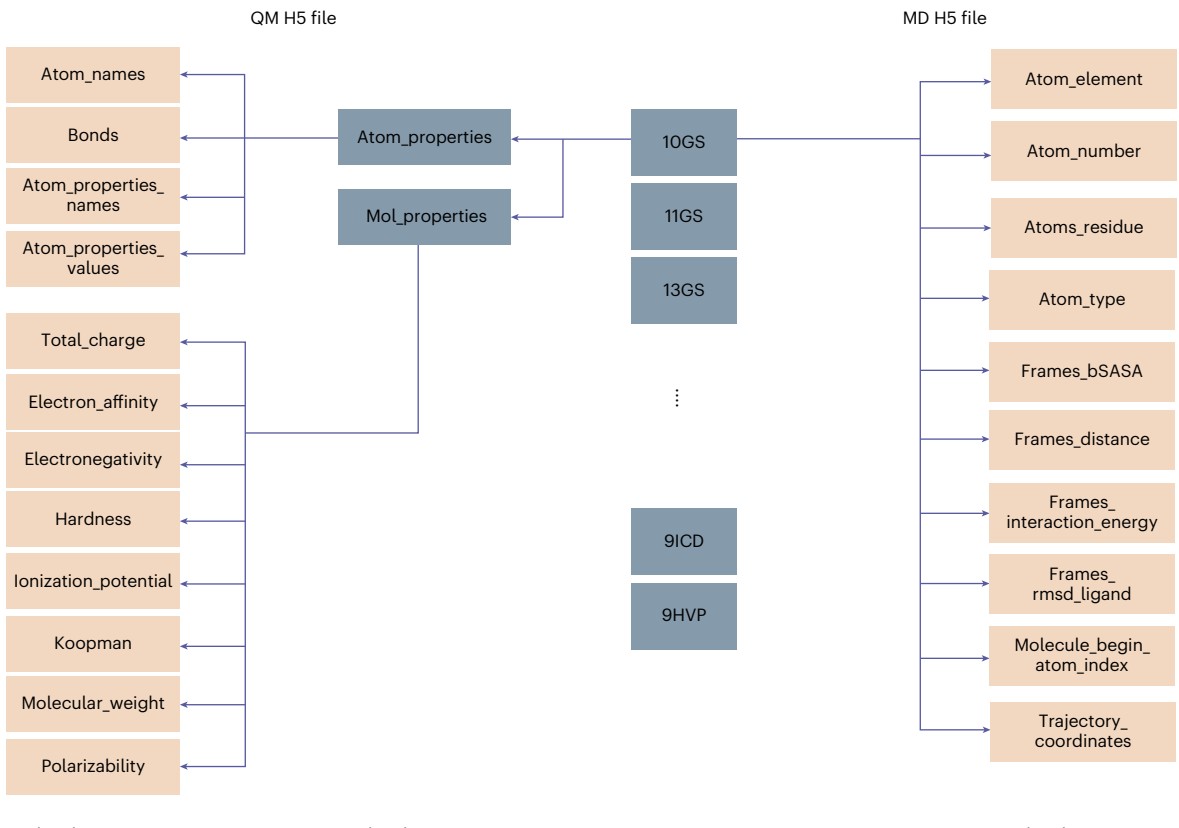

**Fig. 4 | Data hierarchy of the QM and MD files.** The QM data can be accessed via the PDB ID. The properties are split by atom properties and molecular properties. Examples of the calculated molecular properties are given. The electronic densities are provided in a separate file. The MD data are also subdivided by PDB ID. The properties are calculated either for all atoms, for each timestep (frame), or for the whole trajectory, as indicated by the name.

of molecular dipole moments, just like tight-binding Mulliken charges[44]. From the latter, we infer furthermore the reasonableness of the electronic densities provided.

## MD simulations

Experimental structural data are static snapshots that are assumed to represent a thermodynamic most stable state trapped in a crystal but ignore the presence of conformational dynamics. Experimental description of dynamics in biological macromolecules from nanosecond to millisecond timescales is challenging and requires a combination of different spectroscopic techniques. NMR spectroscopy and fluorescence-based methods can provide relevant information but are time consuming, and so far the dynamic information is not well captured in public databases. MD simulations can be performed, starting from experimental structures, and letting them evolve in time using a force field that describes the molecular potential energy surface. Typically, periods of nanoseconds to microseconds can be achieved for individual systems, depending on system size. MD traces allow the analysis of small-range structural fluctuations of the protein–ligand complex, but in some cases large-scale rare events can be observed (Fig. 3). In existing DD software these events are mostly neglected. MD simulations of 16,972 protein–ligand complexes in explicit water were performed for 10 ns. Structures were disregarded whenever non-standard ligand atoms or inconsistencies in the protein starting structures were encountered. A variety of metadata were generated from the simulations to facilitate future AI learning (Fig. 4, Supplementary Table 2 and Supplementary Fig. 1). RMSD$_{Ligand}$ (root-mean-square deviation of the ligand after alignment of the protein) and the root-mean-square deviation of the whole complex were calculated with respect to the native structure. Also, binding affinities were estimated using molecular mechanics

generalized Born surface area (MMGBSA) scoring (no entropic contributions explicitly considered)[45]. Moreover, the buried solvent accessible surface area was obtained for the complex. Calculated properties are stable over the simulations, proving them well equilibrated (Supplementary Fig. 1). For some systems, larger rearrangements of the binding site were captured that in extreme cases led to an opening of the whole binding pocket (Fig. 3). These rare events indicate possible cryptic pockets or transient binding modes. In a small fraction of cases, dissociation was detected (details given in Supplementary Fig. 2).

## AI models

To exemplify possible applications of our dataset, baseline AI models were trained and evaluated. These are included in the repository as a template for future community development. For the QM dataset, the electron affinity and the chemical hardness of the ligand molecules were predicted (Fig. 5). The Pearson correlation is 0.75 for electron affinity and 0.77 for chemical hardness. The mean absolute error shows close predictions to the target values: on average 0.12 eV for electron affinity and 0.13 eV for chemical hardness. For these two exemplary QM features, high accuracy was achieved, opening a route to a fast derivation of QM properties. This is particularly important for larger molecules, where long calculation times are frequent.

For the MD traces, the induced-fit capability of the protein (adaptability) was predicted (see Methods for an exact definition). The model was able to identify elements of biomolecule structure likely to adapt to ligand binding. We achieved a mean Pearson correlation of 0.66. On average 42 of the top 100 atoms were correctly predicted (Fig. 5). As shown in Fig. 5d, the model can predict the atoms in the protein pocket that are mostly flexible during the MD run (large spheres), and detect the more rigid protein regions (small spheres). This allows a fast

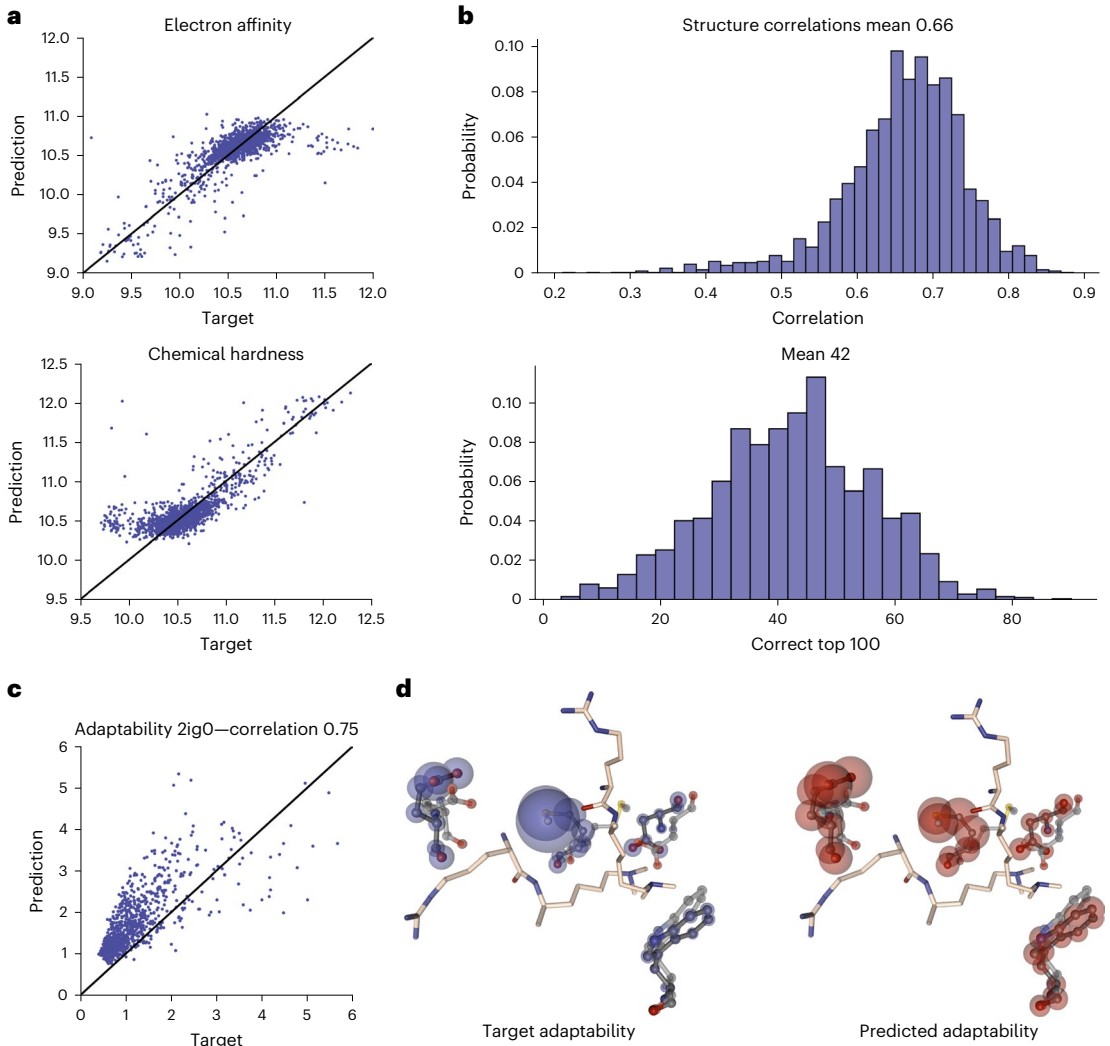

**Fig. 5 | Performance of the AI baseline models. a**, Scatter plot of the predicted against target values of chemical hardness and electron affinity. The AI baseline models to predict QM properties have a high correlation of 0.75 and 0.77 for electron affinity and chemical hardness, respectively. **b**, Adaptability is a measure of the per-atom conformational plasticity of the protein. A histogram of the correlation and the correct top 100 predictions of the adaptability for all structures in the test set are given. An overall mean correlation of 0.66 can be achieved and the mean top 100 accuracy was 0.42 for the adaptability predictions (MD). **c**, Scatter plot for the adaptability result (as in **a**) of example structure 2IG0.

The predicted values are more narrowly distributed than the actual values, but the general trend is correct, as shown by a high correlation value of 0.75. **d**, The adaptability of the residues in the protein pocket highly deviates between the amino acids. The AI model predicts the adaptability given in blue-shaded (target) and red-shaded (AI-predicted) spheres. The radius is scaled according to the adaptability value. The model can correctly identify the rigid residues (small spheres) but also the amino acids with high flexibility. Color and character keys: N, blue, nitrogen; S, yellow, sulfur; O, red, oxygen; C, beige for ligand atoms and black for protein atoms, carbon.

examination of the protein pocket without the necessity of a lengthy MD setup and simulation. The adaptability model gives an innovative example of how experimental structures can be enhanced from the MD-based MISATO data.

A binding affinity AI model combines MISATO MD and QM data. Experimental binding affinities are known to be difficult to compare across different experimental techniques, experimental conditions and calculated affinity types. To decrease these effects, our affinity model predicts a relative affinity of a target structure in relation to a defined base complex. These pairs have the protein and affinity type in common. We achieved high correlations for the MISATO binding affinity benchmark, with improved results using MISATO features when compared with no MISATO features (Fig. 6).

## Experimental validation

The MISATO database and the adaptability AI model were validated on experimental data (Fig. 6). In X-ray crystallography a *B* factor is

determined for each structure in the PDB. It is a measure of the thermal vibration of each atom but usually reflects localized molecular motion as well[46]. We achieved a mean correlation of 0.59 of the *B* factors with the root-mean-square fluctuation (RMSF) in the MISATO MD trajectories. To prove the model against more direct experimental flexibility data, we measured the cap-binding domain of influenza virus polymerase subunit PB2[47] as a model system. Heteronuclear Overhauser effect (hetNOE) NMR measurements, which elucidate flexible protein regions in solution, were performed on this structure (Supplementary Fig. 3). We obtained a high correlation between the calculated adaptabilities and both *B* factors (0.63) and the hetNOE of the protein. A comparison with our adaptability prediction shows that the most flexible regions and the residues of higher rigidity are correctly identified by the model. Quantum chemical methods are required to predict reasonable values for ionization potentials and electron affinities[48]. This applies not only to DFT (density functional theory) but also to ab initio. In Supplementary Data 1 we provide a parameter study performed with

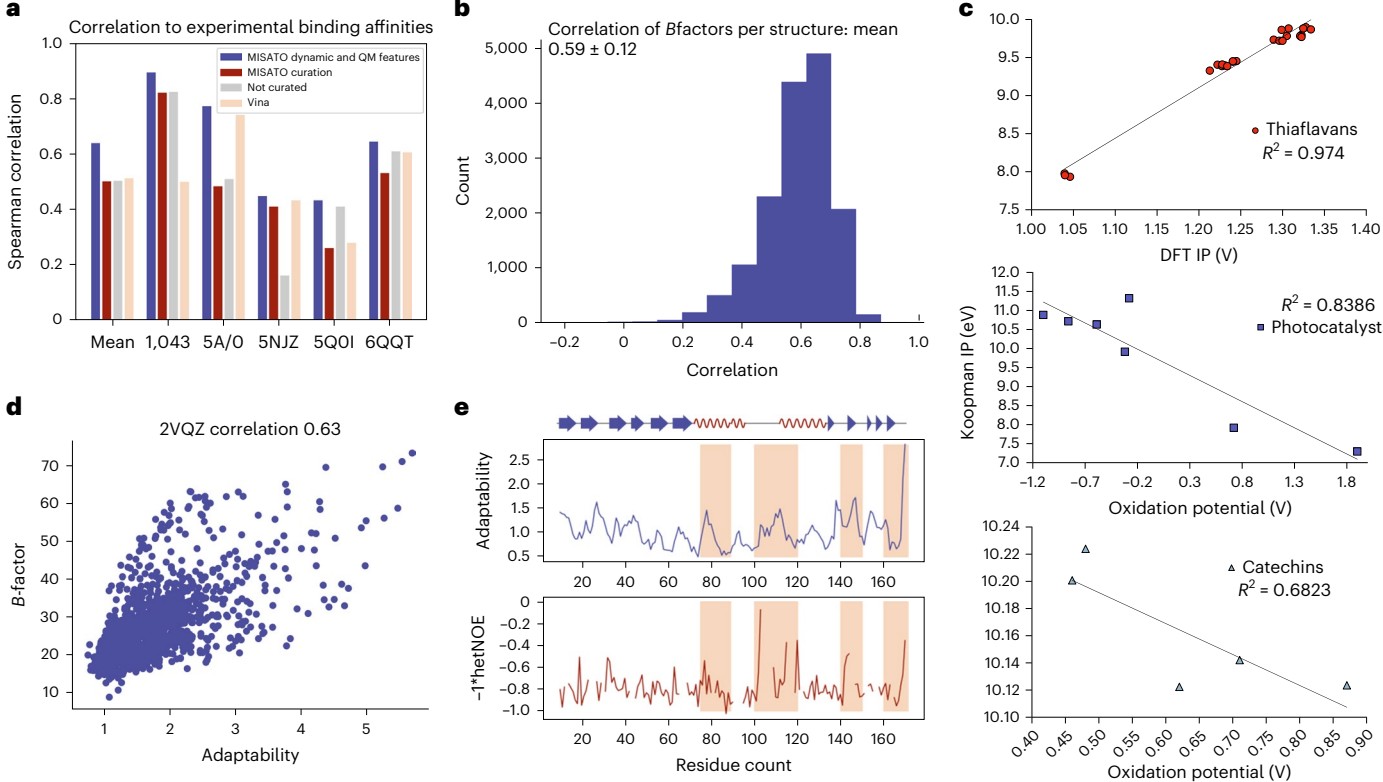

**Fig. 6 | Experimental validation of QM calculations, MD traces and AI models.** **a**, Spearman correlation of the affinity GNN model on the binding affinity benchmark including MISATO features and without features. Moreover, the results using Vina and non-curated complexes (original PDBbind) are shown. We achieved a consistently better performance including QM charges and MD adaptabilities as MISATO features across the affinity benchmark when compared with all other approaches. **b**, Histogram of the correlation of experimental B factors from X-ray crystallography experiments with RMSF calculations from the MD simulations in MISATO. A correlation of 0.59 over all structures was achieved. **c**, High correlations of calculated Koopmans ionization potentials (IP)

from ULYSSES with DFT ionization potentials (upper panel) and experimental oxidation potentials (middle and lower panels) were found for different molecule families. **d**, The cap-binding domain of influenza virus polymerase as a model system for experimental validation of the predicted adaptability. Values given by our AI model had a high correlation of 0.63 against the experimentally determined B factors (which, despite characterizing atom thermal vibration, usually indicates flexibility). **e**, Results of the hetNOE experiments of the cap-binding domain of influenza virus polymerase indicating flexibility of the protein chain were in high accordance with the results of the adaptability model (indicated using shaded regions).

data collected from the CCCBDB database[35], verifying the generality of trends reported in the literature[48,49]. The parameter study shows furthermore that semi-empirical ionization potentials are of a quality similar to, if not higher than, the best DFT results. The advantage, however, is that we systematically apply the same level of theory for all molecules, small and very large alike. We validated the ULYSSES-based calculations of Koopmans ionization potentials against experimental oxidation potentials and DFT-based ionization potentials from the literature for three molecule families. Our calculations correlated highly for photocatalysts (0.84, experimental oxidation potential), catechins (0.68, experimental oxidation potential) and thiaflavans (0.97, calculated DFT data).

**Binding affinity benchmark and validation**

The numerical values describing ligand potency cannot serve as a reliable baseline due to their origin in a wide range of experiments and conditions. These errors in the ground truth cannot be averaged out efficiently. Therefore, we collected high-quality affinity data for 127 ligands for five different protein structures for a MISATO binding affinity benchmark set (Supplementary Table 1 and Supplementary Data 2)[50–54]. This set, being too small for training, can be a reliable validation method for affinity-predicting models. To guarantee reliable affinity data we filtered it to originate from the same publication for each set. Moreover, each of the sets had at least 15 entries with a high dynamical range and few additional occurrences of the protein structure within MISATO.

Our binding affinity graph convolutional network model was evaluated on this benchmark set with and without MISATO features. Additionally, we evaluated the model performance on the original PDBbind set, and using the Vina scoring function. With MD-derived adaptabilities and QM charges, we obtained a mean Spearman correlation of 0.64, which was higher than without the MISATO features (0.50), using Vina (0.51) and using the non-curated database (0.50). Interestingly, an improvement for each of the five sets using the MISATO features could be achieved.

As confirmation of the given results, we evaluated the affinity model on a second benchmark set comprising the six largest clusters of protein structures (clustered on the basis of UniProt ID) of the test set (Supplementary Table 3). These clusters are substantially larger than the sets from the MISATO benchmark and do not necessarily originate from the same publication and the same experimental method within a set. The absolute correlations decreased for this second, more diverse benchmark (Supplementary Fig. 4). Still, we see the same trend as for the first benchmark with a better performance of the MISATO model including adaptability and QM features than the other approaches.

Finally, consistent improvement of affinity prediction model accuracy upon inclusion of QM and dynamic features was observed for the entire curated set as well as selected subsets with high-confidence affinity values (Fig. 6 and Supplementary Figs. 4 and 5). This emphasizes the importance of curation of ligand data and inclusion of at least short-term dynamics in the accuracy of affinity predictions.

The given experiments show that adding the features present in MISATO improves model accuracy over relying on implicit learning from the bare structure.

## Discussion

The great advances over the past years of AI technologies were only possible due to the huge datasets that are fed into these models. In structural biology, the protein folding problem was solved recently, but the DD community still lacks a breakthrough model.

Here, we present MISATO, a database that will open routes in DD for researchers from chemistry, structural biology, biophysics and bioinformatics. MISATO contains the quantum-chemically refined ligand dataset, which permitted the elimination of several structural inaccuracies and crystallographic artifacts. Our refinement protocol can be immediately applied by others for quick database augmentation. We enhance the curated dataset following two orthogonal dimensions. On the one hand, a QM approach supplies systematic electronic properties. On the other hand, a classical approach reveals the system's dynamics and includes the binding affinity and conformational landscape. MISATO contains the largest collection of protein–ligand MD traces to date. Extensive experimental validation of the QM calculations, MD trajectories and AI baseline models highlights the dataset's importance (Fig. 6).

Checkpoint files are made available for potential community extension of the dynamic traces (Supplementary Table 4). Structural biology datasets until now have been unable to incorporate entropy-related information about binding sites and the dynamics of the systems. By conducting MD simulations, it is possible to approximate the conformational space for entropy estimation. A Python interface, built to be intuitively used by anyone, provides preprocessing scripts and template notebooks.

The current limitations of MISATO include the fact that until now the QM calculations were only conducted on the ligand molecules. Moreover, longer timescales of the MD simulations are desirable. These limitations are related to the availability of computing resources. With future releases of MISATO these points will be addressed.

The dataset augmentation presented here paves the way for creative applications of AI models. Our example graph neural network (GNN) model offers quick access to pocket flexibility, a problem never tackled before. This is however just a starting point for a whole class of AI models sprouting from MISATO. Ultimately, we envision models being built on the best of quantum and Newtonian worlds to obtain high-quality thermodynamics, innovatively and efficiently matching the quality of experimental data. With MISATO, AI models will uncover hidden state variables describing protein–ligand complexes.

Altogether, MISATO is meant to provide sufficient training power for accurate, next-generation structure-based DD using AI methods.

## Methods

### Semi-empirical calculations

QM calculations were performed using the ULYSSES library[55], our in-house semi-empirical package. The methods of choice were GFN2-xTB[56], AM1 (ref. 57) and PM6 (ref. 58). Implicit solvation was included using ALPB[59] as parameterized for GFN2-xTB. Selected media included water and wet octanol. Bond orders and hybridizations were estimated using distance-based criteria.

### QM curation of ligand space

Consistent atomic assignments were determined using a series of semi-empirical tests. Semi-empirical quantum chemical methods offer a good compromise between accuracy and computational efficiency[60], which is suitable to refine a collection of almost 20,000 structures of various chemical natures and dimensions (from 6 to almost 370 atoms per molecule). The consistency tests we designed were performed in vacuum to ensure maximum sensitivity of the calculations to structural

inconsistencies. Predicted properties, however, are also obtained using an implicit solvation model.

It is well documented that molecules with many polar groups lack convergence in wavefunction optimization[61]. The same applies when incorrect charges or protonation states are used. Implicit solvation substantially ameliorates the issue and masks problems. In fact, after determining the first guess for total molecular charges, single-point-energy calculations on unrefined ligands using implicit water required roughly 6 h of computation time. Turning off implicit solvation increased the calculation time to almost three weeks on the same machine. This was indicative of severe limitations in proton and total charge assignment. Alternative protonation algorithms were tested—for example, Open Babel[62]. Due to experimental inaccuracies in the geometries, the results were still faulty (Supplementary Figs. 6–9).

Our refinement protocol started with a search for structures with strong atomic overlap. Next, we looked for structures with problematic wavefunction convergence. Vanishing HOMO–LUMO gaps or unpaired electrons flagged further problems, as did violations of the octet rule based on QM population analysis. Finally, we searched for changes in ligand connectivity patterns after QM geometry optimization. This was particularly useful in determining inconsistent protonation states or incorrect electron counting, which generated biradicals. Calculated properties yielded additional testing grounds. Incorrect element assignments were detected when plotting the partial charges against D4 polarizabilities[63] (Fig. 2b).

Severe structural deformations were also detected, inconsistent with the chemical structure (see previous section). For the current stage of the database, we decided to fix only the most extreme cases. This was done using Avogadro (Supplementary Fig. 10)[64]. Further structural refinement is planned.

Whenever our corrections seemed questionable, or the structure was unclear, we checked the original publication. Oxidation states were another sensible point for ligands containing transition metals. Examples of structures we refined are given in the Supplementary Information (Supplementary Figs. 10–12). To ease the inclusion and processing of new structures, a heuristics-based program is included in the database, which performs the basic structural processing (see Supplementary Information for more details). A detailed schematic for the protocol used for cleaning and refining the structures is also given in the Supplementary Information (Supplementary Figs. 13 and 14).

### MD simulations

For all MD simulations, we used the Amber20 (ref. 65) software suite. The protein–ligand complexes were prepared and simulated on the basis of a standard set-up. We parameterized the ligands calculating AM1-BCC[38] charges using antechamber[66] (if the charges did not converge within 1 h we used AM1 charges calculated with ULYSSES). We used the gaff2 (ref. 66) force field for ligands and ff14SB[67] for the proteins. The complexes were neutralized with $Na^+$ and $Cl^-$ ions and solvated in TIP3P[68] explicit water using periodic boundary conditions in an octahedral box (minimum distance between protein and boundary 12 Å).

The complexes were minimized (1,000 steps steepest descent followed by conjugate gradient) and heated to 300 K in several steps within 16 ps. We performed production simulations for 10 ns on all protein–ligand cases in an *NVT* ensemble. The first 2 ns were discarded as equilibration phase, so 8 ns are stored over 100 snapshots for each protein–ligand complex. Using pytraj[31] we calculated different properties of the simulations such as the MMGBSA interaction energy, the buried solvent accessible surface area, the center-of-mass distance between ligand and receptor, and root-mean-square deviations from the native complex.

### Access to the database

The database can be downloaded from Zenodo (Supplementary Table 4). Data are stored in a hierarchical data format. We created two

H5 files, one for the protein–ligand dynamics and one for quantum chemical data, that can be accessed through our container images or after installation of the required Python packages. Installation instructions are given in the repository (Supplementary Table 4). Data are split for each structure using the PDB ID. The feature of interest must also be specified (Fig. 4 and Supplementary Table 2). Python scripts are given in the repository showing how to preprocess the MD dataset for specific cases, only $C_\alpha$ atoms, no hydrogen atoms, only atoms from the binding pocket, and the inclusion of new features. Instructions on how to run inference on new PDB files and visualize the baseline models are given. Checkpoint files for continuing the MD simulations and the electronic densities are provided separately.

## AI applications

For the baseline model for QM predictions, we followed the GNN architecture for small-molecule property prediction in ATOM3D[69]. This model is based on graph convolutions proposed by Kipf and Welling[70] and was adapted for the simultaneous prediction of electron affinity and chemical hardness as essential parameters to describe the ligand. The architecture for the baseline model was a dense layer followed by three sequential layers of NNConv and GRU followed by two dense layers. The model is available via our GitHub repository.

The performance of the ML model was evaluated using correlation and the mean absolute error.

We encode each molecule using the atom positions, the atom type and the bond between the atoms. Each atom corresponds to a node. The atom types are one-hot encoded and edges are defined by selecting the nearest neighbors with a distance of 4.5 Å for each atom. Edges are weighted inversely by the distance between the atoms. We removed outliers straying more than 20 s.d. from the mean values (PDB IDs given in Supplementary Information). All outliers corresponded to molecules containing negatively charged groups and alkyl chains. In other words, these are highly saturated molecules from the electronic viewpoint. Because of their electronic structure, acceptance of an electron is highly unlikely, resulting in very low-to-negative electron affinities. Inaccuracies in the geometries further exacerbate the calculated electron affinities. The results on these systems indicate that some electronic properties are not quantitative; instead, they simply reflect the system's behavior. We trained the GNN with four NVIDIA A100 graphics processing units (GPUs) and 96 CPUs (from 48 physical cores) and for 200 epochs. We used a batch size of 128 and applied a random translation on each node of 0.05 Å.

For the MD task, we modified the GNN architecture from ATOM3D[69] for the node regression task by removing the aggregation of node features into graph features. The architecture for the baseline model was five sequential GCNConv layers[70] followed by two linear layers, summing to 370,000 trainable parameters. The dataset was split into a train (80%), a test (10%) and a validation set (10%) (Supplementary Table 5 and Supplementary Fig. 15) by clustering the amino-acid sequences of the proteins using BlastP[71] to make sure to not have a leakage of similar structural motifs between the splits. We train the GNN with four NVIDIA A100 GPUs and 96 CPUs and for 15 epochs. We use a batch size of eight and a random translation of 0.05 Å. With our model, we calculated the adaptability of each atom during the MD simulation. To this end, we performed an alignment of the coordinates of each simulation with reference to the first frame. To calculate the adaptability $\gamma_x$ for each atom $x$ we take the mean distance of each atom over all timesteps $i$ from the initial position of the atom $\mathbf{r}_{\mathrm{ref},x}$:

$$\gamma_x = \frac{1}{N_{\mathrm{frames}}} \sum_i^{N_{\mathrm{frames}}} |(\mathbf{r}_{\mathrm{ref},x} - \mathbf{r}_{i,x})|.$$

Hydrogen atoms were omitted to reduce the size of the model. For the evaluation, the mean over the results for each structure was calculated. Adaptability gives results very similar to those of RMSF evaluations. We evaluated the performance of our training using Pearson correlation and the average accuracy of the 100 most flexible atoms of each complex.

For the binding affinity task, the data processing, training procedure and GNN architecture were modified. For data processing, all protein–ligand complexes (excluding 1,192 protein–peptide complexes) with known binding affinity were clustered at 30% sequence similarity to avoid data leakage between training (82%), validation (9%) and test (9%) sets (Supplementary Fig. 15). The MISATO affinity benchmark was a holdout part of the test set. Next, clusters were defined on the basis of the UniProt identifier and affinity type, so that each cluster contained only affinity values of the same protein and one of the three affinity types present in the dataset ($K_i$, $K_d$, IC50).

The model predicts the ratio of binding affinities between a pair of protein–ligand complexes. For each cluster, one base molecule was defined that built a pair with each entry of the cluster. The protein–ligand complexes for which no cluster with at least two entries could be defined were discarded (2,259 entries). The atom types were one-hot encoded (omitting hydrogen atoms), and edges were defined following the adaptability model.

One training step consisted of one forward pass for each of the two complexes and mean squared error loss calculation based on the logarithmic ratio of the affinities for each pair. We trained a model including MISATO features and without MISATO features. MISATO atom features comprised calculated adaptabilities (MD) and GFN2-xTB charges in water (QM) for the ligands.

For the GNN architecture, five sequential GCNConv layers were followed by a separate pooling operation for the ligand and protein, respectively. These representations were then further processed via three linear layers with ReLU nonlinearities.

We trained the GNN with four GPUs, 90 CPUs, a batch size of 50 and for 50 epochs. We evaluated the best models on the MISATO affinity benchmark using Spearman correlation on each set.

We used PyTorch v.1.14 to train the models. To code the data loaders and the GNN, we used PyTorch Geometric 2.3.0.

## Scoring of ligands with AutoDock Vina

We calculated an AutoDock Vina[9] score for the MISATO refined protein–ligand complexes of both benchmark sets. We followed a standard preprocessing procedure of generating pdbqt files (see ref. 72 to follow the exact steps). For the receptors we used the prepare_receptor tool on the protonated protein structure from ADFR Suite[73,74]. For the ligands we converted the structures from MOL2 format to pdbqt format using the mk_prepare_ligand.py script. We computed the Vina scores from the generated pdbqt files using the score function from the Python interface of Vina (all scripts can be found via the GitHub page of Vina).

## Correlation with experimental *B* factors

The experimental *B* factors were parsed from published PDB files of crystal structures. For data cleaning, we omitted structures for which 80% of the published *B* factor values had the same entry. Additionally, for some structures, it was not possible to parse the *B* factors correctly due to inconsistencies in the underlying PDB files. The RMSF of each atom of the MD simulation was calculated after superposition to the first frame using pytraj[31].

## Binding affinity benchmark

The benchmark was created by identifying structures in MISATO that originated from the same publication with at least 15 entries. The benchmark was carefully evaluated by assessment of the publication for each of the sets. Only high-quality experimental techniques and data were considered. We further removed sets with a small dynamic range of the affinity data, high coexistence of structures of the same protein within MISATO, and sets with cofactors or metals interacting at the binding site. We obtained a benchmark consisting of five protein sets and 127 bound ligands.

## Protein purification and NMR spectroscopy

The influenza PB2 domain was expressed and purified as previously published[47]. NMR data were acquired at 298 K using a 0.8 mM $^{13}$C-$^{15}$N-PB2 sample on an AV600 spectrometer equipped with a cryoprobe. The sample buffer contained 20 mM sodium phosphate at pH 6.5, and 100 mM NaCl. Standard NMR experiments were used for chemical shift assignments, mainly HNCA, HNCACB, CBCACONH, HNCO, CCONH and HCCH/TOCSY (total correlation spectroscopy). Spectra were processed with the nmrDraw/NMRPipe package[75] and analyzed with NMRView[76].

## Statistics and reproducibility

The splits for train, test and validation were randomized for the different ML models. The exact procedure for each model is given in Supplementary Fig. 15. No statistical method was used to predetermine sample size. For MD, structures were disregarded whenever non-standard ligand atoms (metal ions) or inconsistencies in the protein starting structures were encountered. For the QM model (Supplementary Section 3), a small number (30) of structures were omitted due to the inability of the current algorithm to provide correct predictions for them. This does not introduce a bias to the observation and does not change our observations.

The investigators were not blinded to allocation during experiments or outcome assessment.

### Reporting summary

Further information on research design is available in the Nature Portfolio Reporting Summary linked to this article.

## Data availability

MISATO is publicly accessible and can be downloaded from Zenodo[77] (https://zenodo.org/records/7711953). We provide instructions for usage, data loaders via our GitHub repository, and a container image with all relevant packages installed for GPU usage (Supplementary Table 4). MISATO was built from the PDBbind database (release 2022). Source Data are provided with this paper.

## Code availability

The code can be accessed from our GitHub repository and on Zenodo[78] (https://github.com/t7morgen/misato-dataset). The dataset is accessible via a Python interface using a simple PyTorch data loader. Special attention was given to code modularity, which makes it easy to adjust the AI architecture (Fig. 4 and Supplementary Section 7). We have implemented our dataset according to the ATOM3D[69] code base, a comprehensive suite of ML methods for molecular applications.

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

## Acknowledgements

This work received funding from BMWi ZIM KK 5197901TS0 (T.S., F.M., G.M.P.) and BMBF, SUPREME, 031L0268 (T.S., F.M., G.M.P.). This work was supported by the Helmholtz Association's Initiative and Networking Fund on the HAICORE@FZJ partition.
The funders had no role in study design, data collection and analysis, decision to publish or preparation of the manuscript.

## Author contributions

T.S. and F.M. created and refined the dataset, designed the ML experiments, performed the ML experiments, analyzed the data and wrote the paper. S.B., E.M. and K.D. performed the ML experiments, analyzed the data and contributed to the paper writing. A.S.D.M. performed the NMR experiments and analysis and wrote the NMR section. R.K. selected and validated the affinity benchmark. P.L., S.K., M.P., F.J.T. and M.S. contributed to study design, paper writing and funding of the project. G.M.P. conceived and designed the ML experiments, analyzed the data and wrote the paper.

## Funding

## Competing interests

The authors declare no competing interests.

## Additional information

**Correspondence and requests for materials** should be addressed to Grzegorz M. Popowicz.

# Reporting Summary

## Statistics

For all statistical analyses, confirm that the following items are present in the figure legend, table legend, main text, or Methods section.

| n/a | Confirmed | |
|---|---|---|
| ☐ | ☒ | The exact sample size ($n$) for each experimental group/condition, given as a discrete number and unit of measurement |
| ☒ | ☐ | A statement on whether measurements were taken from distinct samples or whether the same sample was measured repeatedly |
| ☒ | ☐ | The statistical test(s) used AND whether they are one- or two-sided<br>*Only common tests should be described solely by name; describe more complex techniques in the Methods section.* |
| ☒ | ☐ | A description of all covariates tested |
| ☒ | ☐ | A description of any assumptions or corrections, such as tests of normality and adjustment for multiple comparisons |
| ☐ | ☒ | A full description of the statistical parameters including central tendency (e.g. means) or other basic estimates (e.g. regression coefficient) AND variation (e.g. standard deviation) or associated estimates of uncertainty (e.g. confidence intervals) |
| ☒ | ☐ | For null hypothesis testing, the test statistic (e.g. $F$, $t$, $r$) with confidence intervals, effect sizes, degrees of freedom and $P$ value noted<br>*Give P values as exact values whenever suitable.* |
| ☒ | ☐ | For Bayesian analysis, information on the choice of priors and Markov chain Monte Carlo settings |
| ☒ | ☐ | For hierarchical and complex designs, identification of the appropriate level for tests and full reporting of outcomes |
| ☐ | ☒ | Estimates of effect sizes (e.g. Cohen's $d$, Pearson's $r$), indicating how they were calculated |

*Our web collection on statistics for biologists contains articles on many of the points above.*

## Software and code

Policy information about availability of computer code

| Data collection | Amber20, https://gitlab.com/siriius/ulysses version 1.0.0, Python 3.10.9, |
|---|---|
| Data analysis | Python 3.10.9, AutoDock Vina  1.2.5, https://github.com/t7morgen/misato-dataset (doi:10.5281/zenodo.10926008) |

For manuscripts utilizing custom algorithms or software that are central to the research but not yet described in published literature, software must be made available to editors and reviewers. We strongly encourage code deposition in a community repository (e.g. GitHub). See the Nature Portfolio guidelines for submitting code & software for further information.

## Data

Policy information about availability of data

All manuscripts must include a data availability statement. This statement should provide the following information, where applicable:
- Accession codes, unique identifiers, or web links for publicly available datasets
- A description of any restrictions on data availability
- For clinical datasets or third party data, please ensure that the statement adheres to our policy

MISATO is publicly accessible and can be downloaded from Zenodo (https://zenodo.org/records/7711953, DOI: 10.5281/zenodo.7711953). We provide instructions for usage, data loaders via our GitHub repository, and a container image with all relevant packages installed for GPU usage. Source data of Figures 2,3,5,6 is available with this manuscript. MISATO was built from PDBbind database (release 2022).

## Human research participants

Policy information about studies involving human research participants and Sex and Gender in Research.

| | |
|---|---|
| Reporting on sex and gender | Not applicable |
| Population characteristics | Not applicable |
| Recruitment | Not applicable |
| Ethics oversight | Not applicable |

Note that full information on the approval of the study protocol must also be provided in the manuscript.

# Field-specific reporting

Please select the one below that is the best fit for your research. If you are not sure, read the appropriate sections before making your selection.

☒ Life sciences ☐ Behavioural & social sciences ☐ Ecological, evolutionary & environmental sciences

For a reference copy of the document with all sections, see nature.com/documents/nr-reporting-summary-flat.pdf

# Life sciences study design

All studies must disclose on these points even when the disclosure is negative.

| | |
|---|---|
| Sample size | Study was performed on a set of ~20000 published molecular structures. Entire public repository was used for the study. |
| Data exclusions | For MD structures were disregarded whenever non-standard ligand atoms (Metal ions) or inconsistencies in the protein starting structures were encountered. For QM ML model a small number (30) of structures were omitted due to inability of current algorithm to provide correct predictions for them. This does not introduce a bias to the observation and does not change our observations. |
| Replication | Besides NMR no experimental work has been conducted in this manuscript. As a standard, NMR experiments are not performed in replicates. This is because they rely on averaging and Fourier transformation of huge number of experimental samples internally and the measurement error is not arising from single point measurement. The statistical error of NMR experiments is intrinsically captured within one experiment and dominates the error value, the systematic error of NMR measurements is considered very low. |
| Randomization | The splits for train, test and validation were randomized for the different ML models. The exact procedure for each model is given in Supplementary Figure S15. The data for the MD based adaptability model is split into train, validation and test set based on protein sequence similarity. For the QM property model the splits are performed randomly. In case of the affinity model the protein-ligand complexes are first clustered based on UniProt ID. These clusters are then divided into subclusters containing the same affinity type. For each of these subclusters a base molecule is defined and clusters with less than 2 entries are filtered out. The splitting of the clusters into train, test and validation is performed based on sequence similarity as for the adaptability model. The exact splits are available via our GitHub repository. |
| Blinding | No blinding was used in our study. The data we worked with was objective, large and not subject to interpretation or bias. Moreover, we used algorithmic processing of the data that worked uniformly on all data with objective metrics. |

# Reporting for specific materials, systems and methods

We require information from authors about some types of materials, experimental systems and methods used in many studies. Here, indicate whether each material, system or method listed is relevant to your study. If you are not sure if a list item applies to your research, read the appropriate section before selecting a response.

### Materials & experimental systems

| n/a | Involved in the study |
|---|---|
| ☒ | ☐ Antibodies |
| ☒ | ☐ Eukaryotic cell lines |
| ☒ | ☐ Palaeontology and archaeology |
| ☒ | ☐ Animals and other organisms |
| ☒ | ☐ Clinical data |
| ☒ | ☐ Dual use research of concern |

### Methods

| n/a | Involved in the study |
|---|---|
| ☒ | ☐ ChIP-seq |
| ☒ | ☐ Flow cytometry |
| ☒ | ☐ MRI-based neuroimaging |

