## [Peer Review File · Nature Computational Science]

Peer Review Information

Journal: Nature Computational Science

Manuscript Title: MISATO: Machine learning dataset of protein-ligand complexes for structure-based drug discovery

Corresponding author name(s): Dr Grzegorz Popowicz

Editorial Notes:

Reviewer Comments & Decisions:

Decision Letter, initial version:

Date: 11th December 23 15:41:20

Last Sent: 11th December 23 15:41:20

Triggered By: Kaitlin McCardle

From: kaitlin.mccardle@us.nature.com

To: grzegorz.popowicz@helmholtz-munich.de

BCC: kaitlin.mccardle@us.nature.com

Subject: Decision on Nature Computational Science manuscript NATCOMPUTSCI-23-0641A-Z

Message: ** Please ensure you delete the link to your author homepage in this e-mail if you wish to forward it to your co-authors. **

Dear Dr Popowicz,

Your manuscript "MISATO - Machine learning dataset of protein-ligand complexes for structure-based drug discovery" has now been seen by 3 referees, whose comments are appended below. You will see that while they find your work of interest, they have raised points that need to be addressed before we can make a decision on publication.

The referees' reports seem to be quite clear. Naturally, we will need you to address **all** of the points raised.

While we ask you to address all of the points raised, the following points need to be substantially worked on:

- Please be sure to provide a transparent discussion about the experimental data that is reported in the manuscript in order to address comments raised by Reviewer #1
- Please provide quantitative comparisons and experiments as requested by the reviewers.

Please use the following link to submit your revised manuscript and a point-by-point response to the referees' comments (which should be in a separate document to any cover letter):

[REDACTED]

** This url links to your confidential homepage and associated information about manuscripts you may have submitted or be reviewing for us. If you wish to forward this e-mail to co-authors, please delete this link to your homepage first. **

To aid in the review process, we would appreciate it if you could also provide a copy of your manuscript files that indicates your revisions by making use of Track Changes or similar mark-up tools. Please also ensure that all correspondence is marked with your Nature Computational Science reference number in the subject line.

In addition, please make sure to upload a Word Document or LaTeX version of your text, to assist us in the editorial stage.

To improve transparency in authorship, we request that all authors identified as 'corresponding author' on published papers create and link their Open Researcher and Contributor Identifier (ORCID) with their account on the Manuscript Tracking System (MTS), prior to acceptance. ORCID helps the scientific community achieve unambiguous attribution of all scholarly contributions. You can create and link your ORCID from the home page of the MTS by clicking on 'Modify my Springer Nature account'. For more information please visit please visit www.springernature.com/orcid.

We hope to receive your revised paper within three weeks. If you cannot send it within this time, please let us know.

Best regards,

Kaitlin McCardle, PhD
Senior Editor
Nature Computational Science

Reviewers comments:

Reviewer #1 (Remarks to the Author):

The authors have undertaken a thorough computational calculation starting from 19443 protein-ligand structures from pdbBind. In the corresponding reported

structures, incorrect atom assignments and inconsistencies in geometries are not uncommon. The authors have therefore undertaken computationally efficient semi-empirical calculations for the ligands and/or cofactors present in the protein-ligand complexes. Various parameters in terms of geometry-optimization, protonation states, bond-length, bond-angle, have been rectified thereby. The properties of the small molecules further computed with MISATO are electron affinities, chemical hardness, electronegativity, ionization potentials (by definition and using Koopman's theorem), static logP, and polarizabilities. These have been obtained in vacuum, water, and in wet octanol. Atomic properties include partial charges from different models, atomic polarizabilities, bond orders, atomic hybridizations, orbital- and charge-based reactivity (Fukui) indices, and atomic softness. Properties for reactivity indices and softness have also been derived quantum-mechanically. The partial charges computed, for starting classical MD simulations, were considered from HOMO-LUMO level-shift enabling convergence to acceptable electronic states.

From MD simulations protocols, larger arrangements of active sites were observed. With the AI models, the induced fit capability (adaptability of biomolecular structures for ligand binding) has been predicted, depicting target structures with respect to a given base complex. The authors have further predicted the correlation between experimental B-factors and RMSF obtained from simulations. The experimental relative binding affinity correlation has been depicted (in reference to a base structure) with specific case studies as well (using MD derived adaptabilities and QM charges).

There are, however, a few questions/comments.

Two major ones that need to be addressed before this manuscript could be considered for publication - hence suggesting MAJOR REVISION followed by review. The validation part is extremely weak.

* The utility of this dataset depends on how well the calculated binding affinities compare with experimental data. While it is understandable that the experimental data comes from diverse labs/protocols/etc., showing a fair correlation is necessary. The authors have cherry picked very few to show the correlation. They should take all the experimental binding affinities available and compare it with calculated ones.

* Are the quality of binding affinities obtained here better than simple docking calculations? If the docking calculations (takes significantly less human & computational effort) provide better correlations with experimental binding affinities, the whole purpose is lost. The only way to explicitly show this.

Others:

The authors are suggested to provide the details of schematics of the QM calculation protocols (in addition to Fig. 2 b), as per the explanation on the section "QM curation of ligand space", as well as for the AI based prediction protocols for both the training and test datasets, in the Supporting Information.

Since for all the complexes, the QM properties has been refined, it will be easier for the readers to understand in a greater detail a lucid explanation for the AI based QM property prediction (of the test data).

In Figure 6a, the correlation with experimental binding affinity, with and without MISATO features has been provided, and explained in the main text. A separate schematic in the SI, with greater details of the training, test, and validation data, could be more useful for the interpretation of the workflow.

The difference between the MD based AI in predicting the induced fit capability, and the binding affinity prediction protocol illustrated in "AI applications" (and presented in Figure 6), should also be captured in Supporting Information. Perhaps a comparison of the illustrated schematics of both the protocols should be presented in the SI.

The authors however lack the justification in the article, that the accuracy of the MISATO model in predicting AI based binding affinity, will be extrapolated over all other protein-ligand complexes of interest. Calculating the experimental binding affinity (with one of the techniques mentioned in Figure 1, minimizing the experimental limitations as much as possible), for a different (heterogeneous) subset of the MISATO data, might be useful in estimating the correlation between the predicted and experimentally obtained binding affinity, and hence determining the efficacy of the MISATO protocol.

In Figure S8, the heatmap of buried SASA, with distributions around 0 Å², shows higher probability only at the latter end of the simulation time. Please justify the corresponding plot with detailed explanation.

Please ensure there is correct grammatical sentence construction. For example, Page 10, column 2: "To ensure that we have reliable affinity data we selected data that (1) was gathered in the same publication...", this sentence can be better constructed.

Reviewer #2 (Remarks to the Author):

The MISATO database is a well-thought out accumulation of data around binding events for a set of molecules which may be critical for training and evaluating ML models. The breadth of data types made available and conscious attempt to allow the community to expand the MD simulations make this dataset potentially informative for a wide range of experiments and follow up efforts. That being said, there are limitations to the Author's claims around the quality of the dataset on the basis of its performance on the extracted MD features.

Regarding binding affinity prediction, the performance of the GNN with and without MISATO features should be contextualized by the ability for other methods or featurization schemes to predict binding affinity. Without this, it cannot be evaluated whether the 0.64 correlation coefficient is a significant result, or if the improvement from the 0.52 baseline could be accomplished with cheaper methods. Overall, without the context of other methods, these results do not speak one way or another to the appropriateness of the MISATO approach to featurization.

There also an issue with one of the metrics used to assess MD trajectories, "adaptability".

Adaptability calculations are used without providing a definition. I am personally not familiar with it, and by lacking this key understanding, it is hard to interpret the validity of the MD trajectories to extract the relevant information (if adaptability refers to a rare event the 10ns sampling may be insufficient to properly characterize). If adaptability is little more than RMSF or autocorrelation it might be better to use such a well established measure that is intrinsic to the trajectory, rather than introducing a new metric which is suggestive of physical significance that may not be supported by the underlying simulation.

The Authors should either provide an introduction of this metric, together with the information required to assess its validity and the fact that's appropriate to this task, or use a more well-established one that would not need the extra information.

Reviewer #2 (Remarks on code availability):

The repository provides data and code with sufficient amount of information and documentation.

Reviewer #3 (Remarks to the Author):

The authors have developed a large data set of refined protein-ligand (drug-like compounds) complexes based on the already existing PDBbind data base. The PDB bind is just a collection of raw exp. complex structures and the authors convincingly demonstrate that these experimental structures contain many significant errors especially of the bound ligand structures. This is a major imitation to use such data for computational applications in the AI (but also other) areas. The authors use QM methods to "repair" and refine the complexes and evaluate the flexibility and stability using short MD simulations. Finally, on some examples the authors demonstrate how the new data set can be used for AI applications to the drug design task.

In general, I find this a timely, well-designed and very useful effort and important step to design a data set useful for many applications in computational drug design (not only AI). The work has also been carefully controlled and is overall well presented. I have just a few comments:

1. The authors provide an overview on the types of problems encountered when refining bound compounds in Fig. 2. However, I think it is useful for the reader have a general statement in the text how many of the PDBbind complexes need significant refinement and how many could be used directly without QM or MD refinement.
2. Short MD simulations were used to estimate the stability and flexibility of protein-ligand complexes. In Fig. S8 the authors give a general overview on the RMSD and some other properties of the different cases. Also here it is usefull to give more specific information on how many structures show large RMSDs after MD simulation (how many tend to dissociate). Are there types of compounds that show general trends?
3. The authors use the data set to train neuronal networks for predicting some properties of the protein-ligand complexes. I think this very useful to indicate how the data set can be used in AI applications. However, it is also useful to check how these applications work if trained on the original PDBbind data set. Such comparison (at least on one predicted property) could further demonstrate the usefulness of the

refined data compared to the original PDBbind set.

Reviewer #3 (Remarks on code availability):

The github site exists and I downloaded the code and the associated database. I looked at the data and also I tested some of the tasks offered by the code and it worked correctly.

Author Rebuttal to Initial comments

Reviewer #1 (Remarks to the Author):

The authors have undertaken a thorough computational calculation starting from 19443 protein-ligand structures from pdbBind. In the corresponding reported structures, incorrect atom assignments and inconsistencies in geometries are not uncommon. The authors have therefore undertaken computationally efficient semi-empirical calculations for the ligands and/or cofactors present in the protein-ligand complexes. Various parameters in terms of geometry-optimization, protonation states, bond-length, bond-angle, have been rectified thereby. The properties of the small molecules further computed with MISATO are electron affinities, chemical hardness, electronegativity, ionization potentials (by definition and using Koopman's theorem), static logP, and polarizabilities. These have been obtained in vacuum, water, and in wet octanol. Atomic properties include partial charges from different models, atomic polarizabilities, bond orders, atomic hybridizations, orbital- and charge-based reactivity (Fukui) indices, and atomic softness. Properties for reactivity indices and softness have also been derived quantum-mechanically. The partial charges computed, for starting classical MD simulations, were considered from HOMO-LUMO level-shift enabling convergence to acceptable electronic states.

From MD simulations protocols, larger arrangements of active sites were observed. With the AI models, the induced fit capability (adaptability of biomolecular structures for ligand binding) has been predicted, depicting target structures with respect to a given base complex. The authors have further predicted the correlation between experimental B-factors and RMSF obtained from simulations. The experimental relative binding affinity correlation has been depicted (in reference to a base structure) with specific case studies as well (using MD derived adaptabilities and QM charges).

We thank the reviewer for precise analysis, summary and appreciation of the entire scope of our work.

There are, however, a few questions/comments.

Two major ones that need to be addressed before this manuscript could be considered for publication - hence suggesting MAJOR REVISION followed by review. The validation part is extremely weak.

* The utility of this dataset depends on how well the calculated binding affinities compare with experimental data. While it is understandable that the experimental data comes from diverse labs/protocols/etc., showing a fair correlation is necessary. The authors have cherry picked very few to show the correlation. They should take all the experimental binding affinities available and compare it with calculated ones.

We thank the reviewer for this point. While our major objective is to deliver a highly curated dataset of experimental structures with QM and MD extensions to the community, we appreciate the importance of the affinity predictors as one on major tasks such a dataset can be used for.

The MISATO benchmark was constructed so that it had a high dynamical range of affinities in the different sets, so that a meaningful correlation can be calculated. The small numbers of benchmarking molecule originate from the fact that we only selected source publications pending strictest experimental data auditing.

We appreciate the point of the reviewer and calculated the affinities on the whole test set (1083 structures), which comprises the largest possible set that the models did not encounter during training. We could show a clear preference of the model including MISATO

features against the model that did not contain these features. This is despite the obvious deficiencies of experimental data that both the reviewer and us are fully aware of.

The following sentences were added to the results section "Binding affinity benchmark and validation" (page 10):

Lastly, we evaluated the correlation between experimental affinity and predicted affinity for the whole test set (1083 structures, see Fig. S12). We again see a clear preference of the model including the now available MISATO features (0.47) against the model without these features (0.18).

The given experiments show that adding the features present in MISATO improves model accuracy over relying on implicit learning from bare structure.

In the SI the following description of Figure S12 was added:

Figure S12: The affinity model was compared with (left panel) and without (right panel) adaptability and QM features to the corresponding experimental values. The correlation with MISATO features showed a significantly higher performance than without MISATO features.

* Are the quality of binding affinities obtained here better than simple docking calculations? If the docking calculations (takes significantly less human & computational effort) provide better correlations with experimental binding affinities, the whole purpose is lost. The only way to explicitly show this.

We would like to state again that creation of the ultimate scoring function was not the primary objective of our work. However, as suggested by the reviewer we compared the calculated binding affinities to docking scores from AutoDock Vina, one of the most used, evaluated and open-source scoring algorithms. Despite simplicity of the model used, we could confirm that the binding affinity prediction using MISATO features was superior to Vina docking scores.

We added the following sentences to "Binding affinity benchmark and validation" in the results section:

Additionally, we evaluated the model performance on the original pdbBind set and using the Vina scoring function. With MD derived adaptabilities and QM charges we obtained a mean spearman correlation of 0.64 which was higher than without the MISATO features (0.50), using Vina (0.51) and using the not curated database (0.50). Interestingly, an improvement for each of the five sets using the MISATO features could be achieved.

In the description of Fig. 6a we added accordingly:

Moreover, the results using Vina and not curated complexes (original pdbBind) are shown. We achieved a consistently better performance including QM charges and MD adaptabilities as MISATO features across the affinity benchmark compared to all other approaches.

We could confirm the advantage of binding affinity including MISATO features over Vina for a second benchmark set. In "Binding affinity benchmark and validation" we added:

As confirmation of the given results, we evaluated the affinity model on a second benchmark set comprising the six largest clusters of protein structures (clustered based on UniProt id) of the test set (SI Tab. S3). These clusters are significantly larger than the sets from the MISATO benchmark and do not necessarily originate from the same publication and the same experimental method within one set. The absolute correlations decreased for this second, more diverse benchmark (see Fig. S11). Still, we see the same trend as for the first benchmark

with a better performance of the MISATO model including adaptability and QM features than the other approaches.

The results are shown in Fig. S11 with the following sentence referring to the Vina results:

The MISATO affinity including dynamic and QM features (0.43) has a clear improvement in the correlation over the other methods (MISATO curation 0.25, not curated 0.21, Vina 0.37). Interestingly, Vina performed better in relative comparison to the MISATO benchmark and the MISATO curated approach performed slightly better than the original pdbBind (not curated) data.

We added a section in Methods describing the Vina protocol:

Scoring of ligands with AutoDock Vina

We calculated an AutoDock Vina⁹ score for the MISATO refined protein-ligand complexes of both benchmark sets. We followed a standard preprocessing procedure of generating pdbqt files and evaluated the scores using a Vina python interface.

Others:

The authors are suggested to provide the details of schematics of the QM calculation protocols (in addition to Fig. 2 b), as per the explanation on the section "QM curation of ligand space", as well as for the AI based prediction protocols for both the training and test datasets, in the Supporting Information.

Since for all the complexes, the QM properties has been refined, it will be easier for the readers to understand in a greater detail a lucid explanation for the AI based QM property prediction (of the test data).

We fully agree with this point and added a sentence to the main manuscript referring readers to a section in the SI where the QM curation protocols are detailed.

A detailed schematic for the protocol used for cleaning and refining the structures is given in the SI (Fig. S13, S14).

There, we include a schematic representation of our iterative procedure and a full description of the protocol. Since this might be too abstract, at least for readers unfamiliar with (computational) chemistry, we also include a detailed example of a model molecular system that we created to show the main issues we faced during data curation. The description of the protocol then refers the reader to the guided example.

We added Figure S13 and Figure S14 in the SI with the following captions:

Figure S13: Main scheme for the protocol used to clean and refine the original structures.

Figure S14: Example of how the protocol from Figure S13 is applied to an example molecule, created to represent some of the problems faced during database curation.

Additionally, we added a section in the SI with a detailed explanation:

6. Protocol used for database curation

Figures S13 and S14 contain a graphical description of the protocol used for database curation. For simplicity, we will provide a detailed description of Figure S13, referring to S14 when suitable. Note that the protocol is iterative, and it was applied to the whole database,

rather than case by case. The protocol starts by looking for short contacts in the molecules. This can be done using several procedures, namely

- 1) Checking the eigenvalues of the overlap matrix, needed for the quantum mechanical calculations.
- 2) Calculating all the atomic distances for each system, printing all cases where distances are below a given threshold (e.g., 0.6 Angstrom).

In the example of Figure S14, the first iteration shows that two hydrogen atoms are close enough to yield an inconsistency. The least suitable proton has therefore to be removed. After verification that no two atoms are overlapping, the total charge is calculated. Here we used a topological algorithm that detects and identifies specific atomic patterns, associated with functional groups. Each functional group is then given a formal charge, and the summation of all formal charges yields the estimated total charge. Then we verify whether the system contains unpaired electrons (open-shell character). Though in the second iteration there is no problem arising from counting of electrons, the third iteration shows a system with an odd number of electrons. For these cases, manual inspection took place to fix the total charge and the count of electrons. At this stage, electronic density calculations could take place, so we applied convergence analysis. Here we looked mainly at the number of iterations required for having a stable self-consistent field calculation, and we analyzed in detail systems with small HOMO-LUMO gaps, which could indicate pathologies in the atomic system. With a set of converged electronic densities, we performed population analysis, to count the bond-orders between the atoms. Half-bonds (second iteration of Figure S14) or explicit violations of the octet were analyzed in detail, and pathologies fixed. After this check was successful, we assumed that the molecular states were stable and reasonable enough to proceed. We subsequently performed a stability test, where geometry optimization was used to determine whether changes of protonation state, or bond breaking would take place due to the protonation state proposed. Additionally, singlet biradicals could also be identified, as these would lead to an intramolecular reaction. After fixing such inconsistencies, we scouted for outliers in physical properties. An example of one of the tests is provided in Figure 2b of the main text.

Moreover, we added Figure S15 in the SI showing the data flow for the AI based QM model:

Figure S15: Schematic of the data processing workflow for the three baseline models. The exact splits are available via our GitHub repository. **b**, For the QM property model the splits are performed randomly.

We also added two sentences in the section "AI applications" describing the QM model in more detail:

The architecture for the baseline model were a dense layer followed by three sequential layers of NNConv and GRU followed by two dense layers. The model is available via our GitHub repository.

In Figure 6a, the correlation with experimental binding affinity, with and without MISATO features has been provided, and explained in the main text. A separate schematic in the SI, with greater details of the training, test, and validation data, could be more useful for the interpretation of the workflow.

The difference between the MD based AI in predicting the induced fit capability, and the binding affinity prediction protocol illustrated in "AI applications" (and presented in Figure 6), should also be captured in Supporting Information. Perhaps a comparison of the illustrated schematics of both the protocols should be presented in the SI.

We added a schematic to the SI (Figure S15) that makes the data flow of the AI models easier to understand and compare including a detailed description.

Figure S15: Schematic of the data processing workflow for the three baseline models. The exact splits are available via our GitHub repository. **b**, For the QM property model the splits are performed randomly. **c**, In case of the affinity model the protein-ligand complexes are first clustered based on UniProt ID. These clusters are then divided into subclusters containing the same affinity type. For each of these subclusters a base molecule is defined and clusters with less than 2 entries are filtered out. The splitting of the clusters into train, test and validation is performed based on sequence similarity as for the adaptability model.

The authors however lack the justification in the article, that the accuracy of the MISATO model in predicting AI based binding affinity, will be extrapolated over all other protein-ligand complexes of interest. Calculating the experimental binding affinity (with one of the techniques mentioned in Figure 1, minimizing the experimental limitations as much as possible), for a different (heterogeneous) subset of the MISATO data, might be useful in estimating the correlation between the predicted and experimentally obtained binding affinity, and hence determining the efficacy of the MISATO protocol.

The additional benchmarking has been done on the entire test set as described above. We followed the advice of the reviewer and added another benchmark originating from the test set and re-confirmed the results obtained for the first benchmark. We added the following sentences to "Binding affinity benchmark and validation":

As confirmation of the given results, we evaluated the affinity model on a second benchmark set comprising the six largest clusters of protein structures (clustered based on UniProt id) of the test set (SI Tab. S3). These clusters are significantly larger than the sets from the MISATO benchmark and do not necessarily originate from the same publication and the same experimental method within one set. The absolute correlations decreased for this second, more diverse benchmark (see Fig. S11). Still, we see the same trend as for the first benchmark with a better performance of the MISATO model including adaptability and QM features than the other approaches.

The results are given in SI Fig. S11:

Figure S11: Figure as Fig. 6a (main text) but for a different benchmark set. For this benchmark set we chose the six biggest clusters of protein structures (identified through the UniProt identifier). This benchmark is considered harder than the first because the affinity data comprises differing experimental techniques within one set and originates from different publications. Moreover, the sets contain on average a higher number of data points than in the first benchmark. The MISATO affinity including dynamic and QM features (0.43) has a clear improvement in the correlation over the other methods (MISATO curation 0.25, not curated 0.21, Vina 0.37).

A detailed overview of the second benchmark set is provided in SI Table S3:

Table S3: Details of the second benchmark used for the evaluation of Fig. S11.

Set	1	2	3	4	5	6
Number of ligands	342	54	82	44	82	40
Proteins	Kinase	CDK2	Epoxide hydrolase	Src kinase	BTK	MetRS
Range in affinity [kcal/mol]	13.12	13.47	14.83	9.64	11.18	7.13
Example PDB-id	1AQ1	1H1P	1ZD2	2HWO	3GEN	4EG5

In Figure S8, the heatmap of buried SASA, with distributions around 0 \AA^2 , shows higher probability only at the latter end of the simulation time. Please justify the corresponding plot with detailed explanation.

We added another Figure in the SI (S10) complementing the data of Figure S8 and giving a more detailed explanation. We wrote in the caption of Figure S10:

Figure S10: The simulation time in ns is shown against fraction of dissociated structures (see also Fig. S8, lower right panel). We defined a simulation state as dissociated if the COM distance between ligand and receptor was 5 \AA higher in the given snapshot than for the crystal structure. With increasing simulation time, the fraction of dissociated structures increases. The simulation of an entire binding event (unbinding and reassociation) is not possible within 10 ns simulation time, so that only the dissociation of the ligand from the molecule was observed. Overall, a quite low number of 183 dissociation events was tracked, which corresponds to around 0.1 % of the simulations.

Please ensure there is correct grammatical sentence construction. For example, Page 10, column 2: "To ensure that we have reliable affinity data we selected data that (1) was gathered in the same publication...", this sentence can be better constructed.

The mentioned sentence was rewritten:

To guarantee reliable affinity data we filtered it to originate from the same publication for each set. Moreover, each of the sets had at least 15 entries with high dynamical range and little additional occurrences of the protein structure within MISATO.

Reviewer #2 (Remarks to the Author):

The MISATO database is a well-thought out accumulation of data around binding events for a set of molecules which may be critical for training and evaluating ML models. The breadth of data types made available and conscious attempt to allow the community to expand the MD simulations make this dataset potentially informative for a wide range of experiments and follow up efforts. That being said, there are limitations to the Author's claims around the quality of the dataset on the basis of its performance on the extracted MD features.

We thank for the reviewer's insight. Below we specify our efforts to answer the questions and improve the manuscript accordingly.

Regarding binding affinity prediction, the performance of the GNN with and without MISATO features should be contextualized by the ability for other methods or featurization schemes to predict binding affinity. Without this, it cannot be evaluated whether the 0.64 correlation coefficient is a significant result, or if the improvement from the 0.52 baseline could be accomplished with cheaper methods. Overall, without the context of other methods, these results do not speak one way or another to the appropriateness of the MISATO approach to featurization.

The reviewer 2 expressed similar concern as reviewer 1. When creating MISATO dataset, we did not consider affinity prediction benchmark as the main objective. However, following the advice of both reviewers we expanded this part as described in the response to reviewer 1. Moreover, as suggested by the reviewer we contextualized the correlations using AutoDock Vina scores. We could confirm that the binding affinity model including MISATO features has a higher accuracy than using Vina.

For clarity reason we repeat here fragments of the responses to the reviewer 1 as they touch the same subject.

In the results section we added the following sentences to "Binding affinity benchmark and validation":

Additionally, we evaluated the model performance on the original pdbBind set and using the Vina scoring function. With MD derived adaptabilities and QM charges we obtained a mean spearman correlation of 0.64 which was higher than without the MISATO features (0.50), using Vina (0.51) and using the not curated database (0.50). Interestingly, an improvement for each of the five sets using the MISATO features could be achieved.

Additionally, we changed the caption of Fig. 6a:

Moreover, the results using Vina and not curated complexes (original pdbBind) are shown. We achieved a consistently better performance including QM charges and MD adaptabilities as MISATO features across the affinity benchmark compared to all other approaches.

In a second step we could confirm the advantage of binding affinity including MISATO features over Vina for another benchmark set. We added the following sentence to "Binding affinity benchmark and validation":

Still, we see the same trend as for the first benchmark with a better performance of the MISATO model including adaptability and QM features than the other approaches.

In Fig. S11 the Vina results are put into context:

The MISATO affinity including dynamic and QM features (0.43) has a clear improvement in the correlation over the other methods (MISATO curation 0.25, not curated 0.21, Vina 0.37).

Finally, we added a section in Methods describing the protocol used to obtain the Vina scores:

Scoring of ligands with AutoDock Vina

We calculated an AutoDock Vina⁹ score for the MISATO refined protein-ligand complexes of both benchmark sets. We followed a standard preprocessing procedure of generating pdbqt files and evaluated the scores using a Vina python interface.

There also an issue with one of the metrics used to assess MD trajectories, "adaptability". Adaptability calculations are used without providing a definition. I am personally not familiar with it, and by lacking this key understanding, it is hard to interpret the validity of the MD trajectories to extract the relevant information (if adaptability refers to a rare event the 10ns sampling may be insufficient to properly characterize). If adaptability is little more than RMSF or autocorrelation it might be better to use such a well established measure that is intrinsic to the trajectory, rather than introducing a new metric which is suggestive of physical significance that may not be supported by the underlying simulation.

The Authors should either provide an introduction of this metric, together with the information required to assess its validity and the fact that's appropriate to this task, or use a more well-established one that would not need the extra information.

The calculated adaptability is very similar to RMSF with a correlation of around 0.98. We used this terminology to facilitate the readability for the AI, Medicinal Chemistry and Drug Discovery communities.

The adaptability is defined in the section "AI applications" of the Methods section. We added a sentence referring to RMSF:

To calculate the adaptability γ_x for each atom x we take the mean distance of each atom over all timesteps i to the initial position of the atom $\vec{r}_{ref,x}$:

$$\gamma_x = \frac{1}{N_{frames}} \sum_i^{N_{frames}} \left| (\vec{r}_{ref,x} - \vec{r}_{i,x}) \right|$$

Hydrogen atoms were omitted to reduce the size of the model. For the evaluation the mean over the results for each individual structure was calculated. **Adaptability gives very similar results to RMSF evaluations.**

We also added a reference to this definition in the Results section "AI models":

For the MD traces, the induced fit capability of the protein (adaptability) was predicted (**see Methods for an exact definition**).

Reviewer #2 (Remarks on code availability):

The repository provides data and code with sufficient amount of information and documentation.

Reviewer #3 (Remarks to the Author):

The authors have developed a large data set of refined protein-ligand (drug-like compounds) complexes based on the already existing PDBbind data base. The PDB bind is just a collection of raw exp. complex structures and the authors convincingly demonstrate that these experimental structures contain many significant errors especially of the bound ligand structures. This is a major imitation to use such data for computational applications in the AI (but also other) areas. The authors use QM methods to "repair" and refine the complexes and evaluate the flexibility and stability using short MD simulations. Finally, on some examples the authors demonstrate how the new data set can be used for AI applications to the drug design task.

In general, I find this a timely, well-designed and very useful effort and important step to design a data set useful for many applications in computational drug design (not only AI). The work has also been carefully controlled and is overall well presented. I have just a few comments:

We thank the reviewer for appreciation that MISATO is much more than a help in affinity prediction and is meant to improve training for future models of diverse purposes. We have

14

shown that the area of application of our dataset is much broader than, somewhat complex, a matter of docking affinity/score predictors that the other reviewers focused on.

1. The authors provide an overview on the types of problems encountered when refining bound compounds in Fig. 2. However, I think it is useful for the reader have a general statement in the text how many of the PDBbind complexes need significant refinement and how many could be used directly without QM or MD refinement.

We thank the author for this comment. We rewrote the sentence in "Evaluation of QM-based ligand curation" to put more emphasis on this point:

Employing the protocol defined in the previous section we modified a total of 3930 structures, which corresponds roughly to 20 % of the original database **that needed significant refinement** (Fig. 2).

2. Short MD simulations were used to estimate the stability and flexibility of protein-ligand complexes. In Fig. S8 the authors give a general overview on the RMSD and some other properties of the different cases. Also here it is useful to give more specific information on how many structures show large RMSDs after MD simulation (how many tend to dissociate). Are there types of compounds that show general trends?

As suggested by the reviewer we provide a more detailed analysis on the MD simulations with very large RMSDs that tend to dissociate. We added Figure S10 to the SI with an additional analysis of the dissociated ligands and including explicit numbers:

Figure S10: The simulation time in ns is shown against fraction of dissociated structures (see also Fig. S8, lower right panel). We defined a simulation state as dissociated if the COM distance between ligand and receptor was 5 Å higher in the given snapshot than for the crystal structure. With increasing simulation time, the fraction of dissociated structures increases. The simulation of an entire binding event (unbinding and reassociation) is not possible within 10 ns simulation time, so that only the dissociation of the ligand from the molecule was observed. Overall, a quite low number of 183 dissociation events was tracked, which corresponds to around 0.1 % of the simulations.

3. The authors use the data set to train neuronal networks for predicting some properties of the protein-ligand complexes. I think this very useful to indicate how the data set can be used in AI applications. However, it is also useful to check how these applications work if trained on the original PDBbind data set. Such comparison (at least on one predicted property) could further demonstrate the usefulness of the refined data compared to the original PDBbind set.

We consider this a great advice. We included an analysis on the original pdbBind data for the two benchmark sets. On the first benchmark set the predictive performance with and without curation was equally good. On the second benchmark set the curated data performed better. In all these cases the model including the MD and QM derived properties had a significant advantage over the other models. We described these results in the results section "Binding affinity benchmark and validation":

Additionally, we evaluated the model performance on the original pdbBind set and using the Vina scoring function. With MD derived adaptabilities and QM charges we obtained a mean spearman correlation of 0.64 which was higher than without the MISATO features (0.50), using

Vina (0.51) and using the not curated database (0.50). Interestingly, an improvement for each of the five sets using the MISATO features could be achieved.

As confirmation of the given results, we evaluated the affinity model on a second benchmark set comprising the six largest clusters of protein structures (clustered based on UniProt id) of the test set (SI Tab. S3). These clusters are significantly larger than the sets from the MISATO benchmark and do not necessarily originate from the same publication and the same experimental method within one set. The absolute correlations decreased for this second, more diverse benchmark (see Fig. S11). Still, we see the same trend as for the first benchmark with a better performance of the MISATO model including adaptability and QM features than the other approaches.

We also changed the caption of Fig. 6:

We achieved a consistently better performance including QM charges and MD adaptabilities as MISATO features across the affinity benchmark **compared to all other approaches**.

And Figure S11 was added to the SI with the following caption:

Figure S11: Figure as Fig. 6a (main text) but for a different benchmark set. For this benchmark set we chose the six biggest clusters of protein structures (identified through the UniProt identifier). This benchmark is considered harder than the first because the affinity data comprises differing experimental techniques within one set and originates from different publications. Moreover, the sets contain on average a higher number of data points than in the first benchmark. The MISATO affinity including dynamic and QM features (0.43) has a clear improvement in the correlation over the other methods (MISATO curation 0.25, not curated 0.21, Vina 0.37).

Reviewer #3 (Remarks on code availability):

The github site exists and I downloaded the code and the associated database. I looked at the data and also I tested some of the tasks offered by the code and it worked correctly.

Decision Letter, first revision:

Date: 16th February 24 15:11:20

Last Sent: 16th February 24 15:11:20

Triggered By: Kaitlin McCardle

From: kaitlin.mccardle@us.nature.com

To: grzegorz.popowicz@helmholtz-munich.de

BCC: kaitlin.mccardle@us.nature.com

Subject: Decision on Nature Computational Science manuscript NATCOMPUTSCI-23-0641B

Message: ** Please ensure you delete the link to your author homepage in this e-mail if you wish to forward it to your co-authors. **

Dear Dr Popowicz,

Your manuscript "MISATO - Machine learning dataset of protein-ligand complexes for structure-based drug discovery" has now been seen by 3 referees, whose comments are appended below. You will see that while they find your work of interest, they have raised points that need to be addressed before we can make a decision on publication.

The referees' reports seem to be quite clear. Naturally, we will need you to address *all* of the points raised.

While we ask you to address all of the points raised, the following points need to be substantially worked on:

- Please provide overall comparisons for all the (~19 K) datasets with the ground truth.
- Please add additional quantitative validations and ground truth comparisons as requested by Reviewer #1.

Please use the following link to submit your revised manuscript and a point-by-point response to the referees' comments (which should be in a separate document to any cover letter):

[REDACTED]

** This url links to your confidential homepage and associated information about manuscripts you may have submitted or be reviewing for us. If you wish to forward this e-mail to co-authors, please delete this link to your homepage first. **

To aid in the review process, we would appreciate it if you could also provide a copy of your manuscript files that indicates your revisions by making use of Track Changes or similar mark-up tools. Please also ensure that all correspondence is marked with

your Nature Computational Science reference number in the subject line.

In addition, please make sure to upload a Word Document or LaTeX version of your text, to assist us in the editorial stage.

To improve transparency in authorship, we request that all authors identified as 'corresponding author' on published papers create and link their Open Researcher and Contributor Identifier (ORCID) with their account on the Manuscript Tracking System (MTS), prior to acceptance. ORCID helps the scientific community achieve unambiguous attribution of all scholarly contributions. You can create and link your ORCID from the home page of the MTS by clicking on 'Modify my Springer Nature account'. For more information please visit please visit www.springernature.com/orcid.

We hope to receive your revised paper within three weeks. If you cannot send it within this time, please let us know.

Best regards,

Kaitlin McCardle, PhD
Senior Editor
Nature Computational Science

Reviewers comments:

Reviewer #1 (Remarks to the Author):

The authors have not all the queries satisfactorily. The manuscript may be publishable if the following are addressed.

The authors have provided the validation over 1083 test datasets and find it better for MISATO over Vina.

However, they have not provided the overall comparison for all the (~19 K) datasets with the ground truth wherever available.

It is understandable that the technical effort taken by the authors, and with the help of QM calculations, many discrepancies (like boron atom deciphering, instead of bromine atom, in a small molecule) have been pointed out and corrected, which is commendable.

However, the ground truth values of binding free energy reported in the database for those protein-ligand complexes, do not depend on the apparent discrepancies reported for those complexes. The experimental binding affinities are measured before reporting the incorrect protein and/or ligand coordinates/atomic structures and properties.

Along with the detailed analyses by the authors, the absolute value of the correlation

coefficient between the experimental and calculated binding affinity should have been reported along with.

Additionally, see a recent publication (<https://www.nature.com/articles/s41597-023-02872-y>) that has provided the ground truth comparison of the binding affinities for all the protein-ligand complexes.

The authors should elaborate on the ground truth comparison, for all the complexes, after inclusion of the MISATO features.

The representative case studies as shown in Fig. S11 also do not show a consistent increase in correlation coefficient over non-MISATO modifications and Vina.

Further, the absolute number of refinements is 3930, as mentioned in Figure 2, 20% of the total number of complexes taken.

This implies that there is little or no modifications on the remaining selected dataset. This underscores the reason for comparing the whole dataset (including and not including the modified complexes) with the ground truth.

Also, request for a justification of using MMGBSA over MMPBSA in the protocol implemented in the authors' manuscript.

The rest of the queries, especially defining the "adaptability" parameter, and justifying the properties of time dependent buried SASA with additional figures, are justified.

Reviewer #2 (Remarks to the Author):

The authors addressed all the concerns raised in the previous round of reviews.

However, a potentially minor issue still stands: other reviewers have raised some issues regarding the lack of details about the protocol followed for the VINA calculations. The Authors mention:

> We followed a standard preprocessing procedure of generating pdbqt files and evaluated the scores using a Vina python interface.

No references are provided to this extent, but this should be addressed in order to allow others to reproduce and validate the results presented here.

Reviewer #3 (Remarks to the Author):

I carefully read the revised version of the manuscript and also the response to the reviewer comments. I think the authors responded successfully to my concerns and I think the paper is acceptable for publication.

Reviewer #3 (Remarks on code availability):

I was able to download the data and install the code and tested it successfully.

Author Rebuttal, first revision:

Reviewer #1 (Remarks to the Author):

The authors have not all the queries satisfactorily. The manuscript may be publishable if the following are addressed.

The authors have provided the validation over 1083 test datasets and find it better for MISATO over Vina.

However, they have not provided the overall comparison for all the (~19 K) datasets with the ground truth wherever available.

It is understandable that the technical effort taken by the authors, and with the help of QM calculations, many discrepancies (like boron atom deciphering, instead of bromine atom, in a small molecule) have been pointed out and corrected, which is commendable. However, the ground truth values of binding free energy reported in the database for those protein-ligand complexes, do not depend on the apparent discrepancies reported for those complexes. The experimental binding affinities are measured before reporting the incorrect protein and/or ligand coordinates/atomic structures and properties.

Along with the detailed analyses by the authors, the absolute value of the correlation coefficient between the experimental and calculated binding affinity should have been reported along with.

Additionally, see a recent publication (<https://www.nature.com/articles/s41597-023-02872-y>) that has provided the ground truth comparison of the binding affinities for all the protein-ligand complexes.

We thank the reviewer for the input. We included the given reference in the manuscript.

The authors should elaborate on the ground truth comparison, for all the complexes, after inclusion of the MISATO features.

The representative case studies as shown in Fig. S11 also do not show a consistent increase in correlation coefficient over non-MISATO modifications and Vina.

Further, the absolute number of refinements is 3930, as mentioned in Figure 2, 20% of the total number of complexes taken.

This implies that there is little or no modifications on the remaining selected dataset. This underscores the reason for comparing the whole dataset (including and not including the modified complexes) with the ground truth.

We thank the reviewer for this remark. We believe that we have now fully accommodated all raised concerns.

As suggested, we calculated an overall comparison with the ground truth affinity values for all the structures (see Figure S.12). We could show that the correlation was consistently higher using the MISATO features than after not using the MISATO features. We wrote in the SI:

Figure S12: The affinity model was compared on all structures with (left panel) and without (right panel) adaptability and QM features and curation to the corresponding experimental values along the different splits. The correlations to the experimental values are quite similar for each of the splits. Including MISATO features was consistently better (train (grey, 0.52), validation (blue, 0.43), test (red, 0.49), holdout (red, 0.59)) than without the features (train (grey, 0.23), validation (blue, 0.16), test (red, 0.22), holdout (red, 0.38)).

We added in the main text a short description:

Lastly, consistent improvement of affinity prediction model accuracy upon inclusion of QM and dynamic features was observed for entire curated set as well as selected subsets with high-confidence affinity values (Fig. 6, S11 and S12). This emphasizes the importance of curation of ligand data and inclusion of at least short-term dynamics in the accuracy of affinity predictions.

We would like to emphasize that the experimental heterogeneity of affinity data reported and used as the "ground truth" prompts us to give more importance to the small but experimentally consistent sub-datasets than the entire working set. Nevertheless, we observe consistent improvement of affinity prediction model upon inclusion of our curated data.

We could now show across three benchmark sets that the overall performance of MISATO affinity model was higher compared to all other approaches. On a first benchmark, with reliable affinity data, a second benchmark with more diverse affinity data (different experimental methods and origins of the data and higher overall sample number) and a final benchmark including all data available.

Moreover, we have given thorough experimental validation of the dataset using an experimental NMR analysis on a sample case and the experimental B-Factors on all structures available in MISATO.

Also, request for a justification of using MMGBSA over MMPBSA in the protocol implemented in the authors' manuscript.

The MMGBSA and MMPBSA approaches are generally quite similar and only vary in the calculation of the polar solvation part of the interaction energy. The Generalized Born calculation is a very good approximation to the linearized Poisson Boltzmann approach. Across different publications that compare these methods no clear trend is found that favors one over the other. Due to the higher calculation speed and ease of implementation in the pytraj python package we decided to run MMGBSA calculations. The MMPBSA energies can be added in an extension of the database.

The rest of the queries, especially defining the "adaptability" parameter, and justifying the properties of time dependent buried SASA with additional figures, are justified.

We thank the reviewer for the given queries and for the appreciation of our additions to the manuscript.

Reviewer #2 (Remarks to the Author):

22

The authors addressed all the concerns raised in the previous round of reviews.

We thank the reviewer for the acknowledgement of the provided data.

However, a potentially minor issue still stands: other reviewers have raised some issues regarding the lack of details about the protocol followed for the VINA calculations. The Authors mention:

> We followed a standard preprocessing procedure of generating pdbqt files and evaluated the scores using a VINA python interface.

No references are provided to this extent, but this should be addressed in order to allow others to reproduce and validate the results presented here.

As suggested by the reviewer we added more details to the VINA protocol:

We followed a standard preprocessing procedure of generating pdbqt files (see reference⁷⁵ to follow the exact steps). For the receptors we used the prepare_receptor tool on the protonated protein structure from ADFR Suite^{76,77}. For the ligands we converted the structures from MOL2 format to pdbqt format using the mk_prepare_ligand.py script. We computed the VINA scores from the generated pdbqt files using the score function from the python interface of VINA (all scripts can be found via the GitHub page of VINA).

Reviewer #3 (Remarks to the Author):

I carefully read the revised version of the manuscript and also the response to the reviewer comments. I think the authors responded successfully to my concerns and I think the paper is acceptable for publication.

We thank the reviewer for the appreciation of our work and the improvements on the manuscript.

Reviewer #3 (Remarks on code availability):

I was able to download the data and install the code and tested it successfully.

Decision Letter, second revision:

Date: 25th March 24 16:27:53

Last Sent: 25th March 24 16:27:53

Triggered By: Kaitlin McCardle

From: kaitlin.mccardle@us.nature.com

To: grzegorz.popowicz@helmholtz-munich.de

CC: computacionalscience@nature.com

BCC: kaitlin.mccardle@us.nature.com

Subject: AIP Decision on Manuscript NATCOMPUTSCI-23-0641C

Message: Our ref: NATCOMPUTSCI-23-0641C

25th March 2024

Dear Dr. Popowicz,

Thank you for submitting your revised manuscript "MISATO - Machine learning dataset of protein-ligand complexes for structure-based drug discovery" (NATCOMPUTSCI-23-0641C). It has now been seen by the original referees and their comments are below. The reviewers find that the paper has improved in revision, and therefore we'll be happy in principle to publish it in Nature Computational Science, pending minor revisions to satisfy the referees' final requests and to comply with our editorial and formatting guidelines.

TRANSPARENT PEER REVIEW

Nature Computational Science offers a transparent peer review option for original research manuscripts. We encourage increased transparency in peer review by publishing the reviewer comments, author rebuttal letters and editorial decision letters if the authors agree. Such peer review material is made available as a supplementary peer review file. **Please remember to choose, using the manuscript system, whether or not you want to participate in transparent peer review.**

Please note: we allow redactions to authors' rebuttal and reviewer comments in the interest of confidentiality. If you are concerned about the release of confidential data, please let us know specifically what information you would like to have removed. Please note that we cannot incorporate redactions for any other reasons. Reviewer names will be published in the peer review files if the reviewer signed the comments to authors, or if reviewers explicitly agree to release their name. For more information, please refer to our FAQ page.

Thank you again for your interest in Nature Computational Science. Please do not hesitate to contact me if you have any questions.

Sincerely,

Kaitlin McCardle, PhD
Senior Editor
Nature Computational Science

ORCID

Reviewer #1 (Remarks to the Author):

I am satisfied by the follow up analysis and modifications in the manuscript that the authors have done. The manuscript now may be published.

Reviewer #2 (Remarks to the Author):

The Authors have addressed the last concern regarding the lack of information regarding the molecular docking protocol.

However, one of the key references that was added (ref.75) seems to point to a dead link (error 404). Given the importance of that for the reproducibility of the work done here, it would be preferable if this reference could be fixed, possibly with a paper with a proper DOI (that might remain valid for longer than websites).

Final Decision Letter:

Date: 11th April 24 11:16:33

Last Sent: 11th April 24 11:16:33

Triggered By: Kaitlin McCardle

From: kaitlin.mccardle@us.nature.com

To: grzegorz.popowicz@helmholtz-munich.de

BCC: fernando.chirigati@us.nature.com,computationalscience@nature.com,rjsproduction@springernature.com,kaitlin.mccardle@us.nature.com

Subject: Decision on Nature Computational Science manuscript NATCOMPUTSCI-23-0641D

Message Dear Dr Popowicz,

:

We are pleased to inform you that your Resource "MISATO: Machine learning dataset of protein-ligand complexes for structure-based drug discovery" has now been accepted for publication in *Nature Computational Science*.

Once your manuscript is typeset, you will receive an email with a link to choose the appropriate publishing options for your paper and our Author Services team will be in touch regarding any additional information that may be required.

Please note that *Nature Computational Science* is a Transformative Journal (TJ). Authors may publish their research with us through the traditional subscription access route or make their paper immediately open access through payment of an article-processing charge (APC). Authors will not be required to make a final decision about access to their article until it has been accepted. Find out more about Transformative Journals

Acceptance of your manuscript is conditional on all authors' agreement with our publication policies (see <https://www.nature.com/natcomputsci/for-authors>). In particular your manuscript must not be published elsewhere and there must be no announcement of the work to any media outlet until the publication date (the day on which it is uploaded onto our web site).

Before your manuscript is typeset, we will edit the text to ensure it is intelligible to our wide readership and conforms to house style. We look particularly carefully at the titles of all papers to ensure that they are relatively brief and understandable.

Once your manuscript is typeset, you will receive a link to your electronic proof via email with a request to make any corrections within 48 hours. If, when you receive your proof, you cannot meet this deadline, please inform us at rjsproduction@springernature.com immediately.

If you have queries at any point during the production process then please contact the production team at rjsproduction@springernature.com.

You may wish to make your media relations office aware of your accepted publication, in case they consider it appropriate to organize some internal or external publicity. Once your paper has been scheduled you will receive an email confirming the publication details. This is normally 3-4 working days in advance of publication. If you need additional notice of the date and time of publication, please let the production team know when you receive the

proof of your article to ensure there is sufficient time to coordinate. Further information on our embargo policies can be found here:
<https://www.nature.com/authors/policies/embargo.html>

We welcome the submission of potential cover material (including a short caption of around 40 words) related to your manuscript; suggestions should be sent to Nature Computational Science as electronic files (the image should be 300 dpi at 210 x 297 mm in either TIFF or JPEG format). We also welcome suggestions for the Hero Image, which appears at the top of our home page; these should be 72 dpi at 1400 x 400 pixels in JPEG format. Please note that such pictures should be selected more for their aesthetic appeal than for their scientific content, and that colour images work better than black and white or grayscale images. Please do not try to design a cover with the Nature Computational Science logo etc., and please do not submit composites of images related to your work. I am sure you will understand that we cannot make any promise as to whether any of your suggestions might be selected for the cover of the journal.

Best regards,

Kaitlin McCardle, PhD
Senior Editor
Nature Computational Science

P.S. Click on the following link if you would like to recommend Nature Computational Science to your librarian: <https://www.springernature.com/gp/librarians/recommend-to-your-library>

** Visit the Springer Nature Editorial and Publishing website at www.springernature.com/editorial-and-publishing-jobs for more information about our career opportunities. If you have any questions please click here.**